

# Interannual variability of summertime formaldehyde (HCHO) vertical column density and its main drivers in northern high latitudes

Tianlang Zhao[1], Jingqiu Mao[1], Zolal Ayazpour[2,3], Gonzalo González Abad[3], Caroline R

Nowlan[3], Yiqi Zheng[1]

[1] University of Alaska Fairbanks, Department of Chemistry and Biochemistry & Geophysical

Institute, Fairbanks, AK, United States

[2] University at Buffalo, Department of Civil, Structural and Environmental Engineering, Buffalo,

NY, United States

[3] Center for Astrophysics, Harvard & Smithsonian, Cambridge, MA, United States

*Correspondence to*: Tianlang Zhao (tzhao@alaska.edu) and Jingqiu Mao (jmao2@alaska.edu)

**Abstract:** The northern high latitudes (50-90°N, mostly including boreal forest and tundra

ecosystem) has been undergoing rapid climate and ecological changes over recent decades, leading

to significant variations in volatile organic compounds (VOCs) emissions from biogenic and

biomass burning sources. HCHO, a widely used indicator of VOC emission, exhibits high climate

sensitivity. However, the interannual variability of HCHO and its main drivers over the region

remain unclear. In this study, we use the GEOS-Chem chemical transport model and satellite

retrievals from Ozone Monitoring Instrument (OMI) and Ozone Mapping and Profiler Suite

(OMPS) to examine HCHO vertical column density (VCD) interannual variations in summertime

of 2005-2019. Our results show that wildfires heavily influence interannual variability of HCHO VCD over Siberia, Alaska, and North Canada, while biogenic emissions and background methane oxidation are the predominant drivers of HCHO interannual variability over East Europe. Solar-induced chlorophyll fluorescence (SIF) from Orbiting Carbon Observatory-2 (OCO-2) provides additional evaluation for HCHO interannual variability from biogenic emission, showing potential of constraining biogenic emission in northern high latitudes.

### 1. Introduction

VOCs are main precursors of tropospheric ozone and secondary organic aerosols, strongly impacting air quality and climate (Jin et al., 2017; Mao et al., 2018; Jin et al., 2020; Zheng et al., 2020). HCHO is mainly produced from atmospheric VOC oxidation with a short photochemical lifetime on the order of hours, serving as an indicator of non-methane VOC (NMVOC) emissions and photochemical processes (Fu et al., 2007; Millet et al., 2008). Understanding the interannual variability of HCHO is important for quantifying long-term trend of VOC emissions in response to climate changes and air quality control implementation.

Several studies suggest that biogenic VOC emissions are largely responsible for interannual variabilities of HCHO on a global scale (Palmer et al., 2001; De Smedt et al., 2008; González Abad et al., 2015; De Smedt et al., 2018). Stavrakou et al. (2009) attributes Biogenic VOCs (BVOCs) emissions as the predominant source of global HCHO columns, in which isoprene alone contributes 30% of global HCHO, according to model estimation. Isoprene emissions were also found to be the major driver of HCHO interannual variability (Bauwens et al., 2016; Stavrakou et al., 2018; Morfopoulos et al., 2022). Pyrogenic and anthropogenic sources can also contribute to a

large part of HCHO in regional scales, such as in the Amazon (Zhang et al., 2019) and Alaska (Zhao et al., 2022).

Northern high latitudes are experiencing rapid Arctic warming in recent decades, resulting in strong increases in BVOC emissions (Lappalainen et al., 2009; Vedel-Petersen et al., 2015; Kramshøj et al., 2016; Seco et al., 2022). Several studies suggest monoterpenes to be the predominant BVOC species in boreal forests over middle and north Europe,and southeastern

Siberia (Rinne et al., 2000; Spirig et al., 2004; Timkovsky et al., 2010; Bäck et al., 2012; Rantala et al., 2015; Juráň et al., 2017; Zhou et al., 2017). This BVOC speciation appears to be different in the boreal forests in Alaska, North Canada and East Siberia, where isoprene appears to be the predominant BVOC species (Blake et al., 1992; Timkovsky et al., 2010; Zhao et al., 2022). BVOC measurements in tundra system show a very strong positive temperature dependence for isoprene

fluxes, over Greenland (Vedel-Petersen et al., 2015; Kramshøj et al., 2016; Lindwall et al., 2016a), northern Sweden (Faubert et al., 2010; Tang et al., 2016) and the Alaskan North Slope (Potosnak et al., 2013; Angot et al., 2020; Selimovic et al., 2022). Whether or not these changes can be seen from satellite HCHO observations remains unclear.

Wildfire is another major source of VOCs with large direct emissions of HCHO (Permar et al., 2021). A number of studies have shown positive trend and strong interannual variability of wildfires over Arctic regions in the past few decades (Kelly et al., 2013; Giglio et al., 2013; Descals et al., 2022), suggesting an increasingly important role of wildfires on HCHO sources. Several modelling studies suggest that wildfires can become the main source of HCHO over Alaska (Zhao

et al., 2022), Siberia and Canada (Stavrakou et al., 2018). In fact, the contribution from wildfires

could be even larger as models tend to underestimate the secondary production of HCHO from other VOC precursors (Alvarado et al., 2020; Zhao et al., 2022; Jin et al., 2023). To what extent wildfires contribute to HCHO interannual variability remains unclear.

Satellite Solar Induced Fluorescence (SIF) could potentially provide additional constraints on biogenic-related HCHO column over northern high latitudes, due to their similar dependence on temperature and light availability (Foster et al., 2014; Zheng et al., 2015). SIF is the re-emission of light by plants as a result of absorbing solar radiation during photosynthesis and is widely used to estimate vegetation productivity and health (Porcar-Castell et al., 2014; Magney et al., 2019).

Isotopic labeling studies show that 70-90% of isoprene production is from chloroplasts, directly linked to photosynthesis (Delwiche and Sharkey, 1993; Karl et al., 2002; Affek and Yakir, 2003). As SIF is directly linked to flux-derived Gross Primary Productivity (GPP) and HCHO can be largely explained by isoprene emissions (Zheng et al., 2017), we expect some correlation between satellite observations of HCHO and SIF under certain conditions.


Here we use newly retrieved HCHO vertical column density (VCD) products from OMI and OMPS, combined with GEOS-Chem chemical transport model, to examine summertime HCHO spatiotemporal variability over northern high latitudes from 2005 to 2019. The satellites and the model are introduced in section 2. In section 3, we evaluate spatial and interannual variability of

modelled HCHO VCD using OMI and OMPS retrievals. Section 4 presents model sensitivity tests, demonstrating how background HCHO, wildfire and biogenic VOC emissions influence HCHO interannual variability across Alaska, Siberia, North Canada and East Europe in northern high



latitudes. In section 5, we evaluate biogenic HCHO interannual variability using satellite SIF data. Summary and discussion are in section 6.

## 2. Observations and Model

### 2.1. Observational data sets

We use satellite observations of tropospheric HCHO columns from OMI and OMPS to evaluate summertime HCHO variability in northern high latitudes. OMI is a UV/Visible backscatter spectrometer on-board the Aura satellite launched in July 2004, with global daily coverage at an overpass time of 13:30 LT. OMPS is a spectrometer on-board two satellites: NASA/NOAA SUOMI NPP (hereafter SNPP) and NOAA-20, which launched in October 2011 and November 2017, respectively. The nadir spatial resolution of OMI and OMPS-SNPP are $13\times24$ km$^2$ and $50\times50$ km$^2$ respectively (de Graaf et al., 2016; Levelt et al., 2006). Here we use OMI data from 2005 to 2019 and OMPS-SNPP data from 2012 to 2019. We calculate monthly mean HCHO VCD from OMI HCHO VCD retrieval (OMHCHO Version-4) product (González Abad et al., 2022) and OMPS-SNPP Level 2 HCHO total column V1 product (Nowlan et al., 2023) provided by the Smithsonian Astrophysical Observatory.

The OMI and OMPS HCHO retrievals use a three-step procedure to calculate the HCHO VCD (Nowlan et al., 2023). First, the slant column density (SCD) is determined through spectral fitting of a backscattered radiance spectrum collected in the wavelength region of 328.5 to 356.5 nm. This fit uses a daily reference spectrum (one for each cross-track position) determined from radiances collected over a relatively clean area of the Pacific between latitudes 30°S and 30°N.



The area used for this reference calculation is referred to as the reference sector. Second, scene-by-scene radiative transfer calculations are performed to determine vertically-resolved scattering weights, which can be used to determine the air mass factor (AMF) in combination with the trace gas profile (Palmer et al., 2001). This AMF describes the path of light and is used for converting the SCD to a VCD (VCD=SCD/AMF). Third, the background reference slant column ($SCD_R$) in the radiance sector region is determined using a model, to correct the retrieved SCD which is in fact the differential SCD determined from the ratio of the observed radiance and the reference radiance. A further bias correction ($SCD_B$) is applied to reduce high-latitude biases, which mostly affect OMPS-SNPP (Nowlan et al., 2023).

To compare with modelled results, OMI and OMPS-SNPP HCHO retrievals are reprocessed following a three-step procedure. This is primarily done to replace the climatologies used in the OMI and OMPS-SNPP products with our own GEOS-Chem simulations. First, we remove the data points falling in the following criteria: (1) main quality flag > 0, (2) cloud cover fraction $\geq$ 40%, (3) solar zenith angle (SZA) $\geq$ 70°, and (4) Ice/snow flag = 1. After filtering, we regrid the level 2 swath data in the local time window 12:00–15:00 LT to 0.5°×0.625° horizontal resolution. Second, we calculate the air mass factor ($AMF_{GC}$) based on local GEOS-Chem HCHO vertical profile and satellite scattering weight (Palmer et al., 2001). Third, we calculate the slant column density of HCHO in the reference sector ($SCD_{R,SAT}$), using modelled HCHO reference sector column and satellite air mass factor over the same location ($VCD_{R,GC}$ and $AMF_{R,SAT}$) (De Smedt et al., 2018; Zhu et al., 2016):

$$SCD_{R,SAT} = VCD_{R,GC} \times AMF_{R,SAT} \qquad (1)$$



VCD$_{R,GC}$ is calculated by global monthly climatology of hourly HCHO profiles at the time of overpass, from a 2018 GEOS-Chem high-performance (GCHP) run at 0.5°×0.5° resolution (Bindle et al., 2021; Eastham et al., 2018). AMF$_{R,SAT}$ is the AMF from the satellite product, which is calculated using the VLIDORT radiative transfer model as described in Nowlan et al. (2023). We rearrange the satellite vertical column as following:

$$VCD_{SAT,reprocessed} = (\Delta SCD_{SAT} + SCD_{B,SAT} + SCD_{R,SAT}) \, / \, AMF_{GC} \qquad (2)$$

Here $\Delta SCD_{SAT}$ is the fitted HCHO slant column, $SCD_{B,SAT}$ is the bias correction term for unexplained background patterns in the HCHO retrievals which may be due to instrument or retrieval issues (Nowlan et al., 2023). The single-scene precision of the retrieval is $1\times10^{16}$ molecules cm$^{-2}$ (absolute) for OMI and $3.5\times10^{15}$ molecules cm$^{-2}$ for OMPS-SNPP from spectral fitting and 45–105% (relative) from the AMF (González Abad et al., 2015; Nowlan et al., 2023). The spectral fitting error is primarily random in individual measurements, while the AMF error has both random and systematic components. The precision can be improved by spatial and temporal averaging (De Smedt et al., 2008; Zhu et al., 2016; Boeke et al., 2011). Our analyses in this work are based on monthly data, so the absolute uncertainty in HCHO column is reduced to $<1\times10^{15}$ molecules cm$^{-2}$ (De Smedt et al., 2018).

We utilize high-resolution SIF estimates derived from OCO-2 and MODIS (doi: https://doi.org/10.3334/ORNLDAAC/1863, last accessed: August 10, 2022). These datasets provided globally contiguous daily SIF estimates at a spatial resolution of approximately 0.05° ×

0.05° (around 5 km at the equator) and a temporal resolution of 16 days, from September 2014 to

July 2020. The dataset was estimated by using an Artificial Neural Network (ANN) trained on the

native OCO-2 SIF observations and MODIS BRDF-corrected seven-band surface reflectance

along orbits of OCO-2. The ANN model was subsequently used to predict daily mean SIF (mW

$m^{-2}nm^{-1}sr^{-1}$) in the gap regions based on MODIS reflectance and land cover. In our study, the OCO-

2 SIF estimates are monthly averaged and regridded to 0.1° × 0.1° spatial resolution for the

comparison with OMI HCHO VCD, and regridded to 2° × 2.5° spatial resolution when comparing

with GEOS-Chem results.

### 2.2. Global GEOS-Chem simulations

GEOS-Chem, a 3-D global chemical transport model, is used in this study to examine the

spatiotemporal variability of HCHO and VOCs across northern high latitudes (50-90°N). The

model is driven by the Modern-Era Retrospective analysis for Research and Applications, Version

2 (MERRA-2), provided by the Global Modeling and Assimilation Office (GMAO) at NASA's

Goddard Space Flight Center (Rienecker et al., 2011). GEOS-Chem version 12.7.2 is employed

(http://wiki.seas.harvard.edu/geos-chem/index.php/GEOS-Chem_12#12.7.2, last access: August

10, 2022) with an update on cloud chemistry (https://github.com/geoschem/geos-chem/issues/906,

last access: August 10, 2022). The simulations encompass 15 summers (1 May to 31 August) from

2005 to 2019, at a horizontal resolution of 2°×2.5° and 72 vertical layers from the surface to

0.01 hPa.

Biomass burning emissions in our simulation are derived from the Global Fire Emission Database

(GFED4.1s) inventory (Giglio et al., 2013). Year-specific GFED4.1s inventory is used in each

year of the simulation to make sure the representation of the interannual variability in wildfire

emissions. Emissions on a 3-hour basis are obtained from MODIS satellite observations, which

provide information on fire detection and burning area (van der Werf et al., 2017). The GFED4.1s

inventory reports the HCHO emission factor of 1.86 g/kg dry matter for boreal forest fires and

2.09 g/kg dry matter for temperate forest fires, aligning with recent field measurements (Liu et al.,

2017; Permar et al., 2021).

BVOC emissions in the study are calculated online (Emission factor maps computed online) using

the Model of Emissions of Gases and Aerosols from Nature (MEGAN, v2.1) (Guenther et al.,

2006, 2012). Terrestrial vegetation for BVOC emissions is based on the plant functional type

(PFT) distribution derived from Community Land Model (CLM4) (Lawrence et al., 2011; Oleson

et al., 2013). CLM4 output (Figure S1) suggests two major PFTs over northern high latitudes:

broadleaf deciduous boreal shrubs (mainly over the north and south Alaska, North Canada and

North Siberia) and needle leaf evergreen boreal trees (mainly over interior Alaska, North Canada,

south Siberia and the northern part of East Europe), both with high emission factors in isoprene

and low emission factors in monoterpenes. The southern part of East Europe is dominated by

croplands and broadleaf deciduous temperate trees. In this work, 'monoterpenes' from model

calculation include $\alpha$-pinene, $\beta$-pinene, sabinene and carene.


In this study, we use the detailed $O_3$-$NO_x$-$HO_x$-VOC chemistry ("tropchem" mechanism) (Park et

al., 2004; Mao et al., 2010, 2013), incorporating updates on isoprene chemistry (Fisher et al.,

2016). The performance of this version of isoprene chemistry in GEOS-Chem has been extensively

evaluated using recent field campaigns and satellite observations over the southeast US (Fisher et



al., 2016; Travis et al., 2016), including HCHO production from isoprene oxidation (Zhu et al.,

2016, 2020; Kaiser et al., 2018). The ability of GEOS-Chem with this chemistry to reproduce the

vertical profiles of HCHO observed during the Alaska summer, as shown in the ATom-1 in-situ

campaign, has been demonstrated (Zhao et al., 2022). Under high-$NO_x$ conditions (1 ppbv), HCHO

production is rapid, reaching 70-80% of its maximum yield within a few hours, whereas under

low-$NO_x$ conditions (0.1 ppbv or lower), it takes several days to reach the maximum yield, and the

cumulative yield is approximately 2-3 times lower than that under high-$NO_x$ conditions (Marais et

al., 2012).

To examine the influence of different sources on HCHO columns in northern high latitudes, we

conducted a series of GEOS-Chem simulations, as is described in Table 1, to separate modelled

HCHO total column ($VCD_{GC}$) into three parts: the background column ($VCD_{0,GC}$), biogenic

emission induced column ($dVCD_{Bio,GC}$) as well as wildfire emission induced column

($dVCD_{Fire,GC}$), assuming they are independent:

$$VCD_{GC} = VCD_{0,GC} + dVCD_{Bio,GC} + dVCD_{Fire,GC} \quad (3)$$

$VCD_{0,GC}$ is the $VCD_{GC}$ from the GEOS-Chem simulation in which both biogenic and wildfire

emissions are turned off. $VCD_{0,GC}$, $dVCD_{Fire,GC}$ and $dVCD_{Bio,GC}$ are derived by:

$$VCD_{0,GC} = VCD_{GC}(BG) \qquad (4a)$$

$$dVCD_{Fire,GC} = VCD_{GC}(All) - VCD_{GC}(NF) \qquad (4b)$$

$$dVCD_{Bio,GC} = VCD_{GC}(NF) - VCD_{0,GC} \qquad (4c)$$





In Figure 1(a) we display the extent of four domains focused on in this work. The selection of

Alaska domain follows Zhao et al (2022); East Europe and Siberia domains follow Bauwens et al

(2016); northern Canada domain follows the North America domain in Bauwens et al (2016) but

excluded Alaska.

To emphasize the key drivers of HCHO interannual variability, we classify 2005-2019 into "high

HCHO years" and "low HCHO years" for each of four domains. For each domain, the years that

have above-average May-August sum of regional-averaged monthly $VCD_{GC}$ will be classified as

"high HCHO year"; those years have the value below average will be classified as "low HCHO

years". The classification is shown in Table S1.

**Table 1. Configuration of GEOS-Chem global simulations in this study**

| Simulations | Biogenic emission | Wildfire |
|---|---|---|
| **Biogenic + wildfire + Background (All)** | On | On |
| **Background (BG)** | Off | Off |
| **Biogenic + Background (NF)** | On | Off |

We use the coefficient of variation (CV) to quantify the interannual variability of summertime

HCHO. CV is defined as the ratio of the standard deviation to the mean (CV $= \frac{\sigma}{\mu}$), which is a

measure of interannual variability (Giglio et al., 2013). Assuming $VCD_{0,GC}$, $dVCD_{Bio,GC}$,

$dVCD_{Fire,GC}$ are three independent components of $VCD_{GC}$, we have $\sigma^2_{VCD_{GC}} = \sigma^2_{VCD_{0,GC}} +$





$\sigma^2_{dVCD_{Bio,GC}} + \sigma^2_{dVCD_{Fire,GC}}$, so the contribution of each component to the CV of VCD$_{GC}$ can be calculated by:

$$CVcontribution_{VCD_{0,GC}} = \frac{\sigma^2_{VCD_{0,GC}}}{\sigma^2_{VCD_{GC}}} \qquad (5a)$$

$$CVcontribution_{dVCD_{Bio,GC}} = \frac{\sigma^2_{dVCD_{Bio,GC}}}{\sigma^2_{VCD_{GC}}} \qquad (5b)$$


$$CVcontribution_{dVCD_{Fire,GC}} = \frac{\sigma^2_{dVCD_{Fire,GC}}}{\sigma^2_{VCD_{GC}}} \qquad (5c)$$

### 3. OMI/OMPS Evaluation with GEOS-Chem HCHO VCD

Figure 1 shows the July mean HCHO VCD over northern high latitudes during 2012-2019, from reprocessed OMI, OMPS-SNPP and GEOS-Chem. We show that OMI and OMPS-SNPP HCHO VCD have consistent spatial pattern and their magnitude agree within 15% (Panel a, b and d).

OMPS-SNPP does show lower values in some regions, perhaps due to several cloud and surface reflectance assumptions made in OMPS-SNPP retrievals, or biases that may persist at high-latitudes and large solar zenith angles (Nowlan et al., 2023). While GEOS-Chem well reproduced the spatial pattern of HCHO VCD that OMI and OMPS-SNPP captured (Panel c), we find that GEOS-Chem HCHO VCD is lower than that of OMI by 40%, particularly over wildfire impacted

areas (Panel e). The model-satellite discrepancies in wildfire areas can be in part due to model underestimates of VOC emissions and HCHO production from wildfire plumes (Jin et al., 2023), and in part due to the large uncertainties associated with satellite retrievals in the presence of wildfire smokes (Jung et al., 2019). The model-satellite discrepancies outside wildfire areas can be also due to biases in both model and satellites. For example, Stavrakou et al (2015) attributed

the discrepancy at northern high latitudes to low bias in isoprene emissions. Recent studies suggest



that TROPOMI HCHO retrieval may have a positive bias under low HCHO conditions (Vigouroux et al., 2020). OMPS-SNPP HCHO shows a similar positive bias at clean sites, but has a closer agreement with FTIR HCHO columns at polluted sites (Nowlan et al., 2023; Kwon et al., 2023).


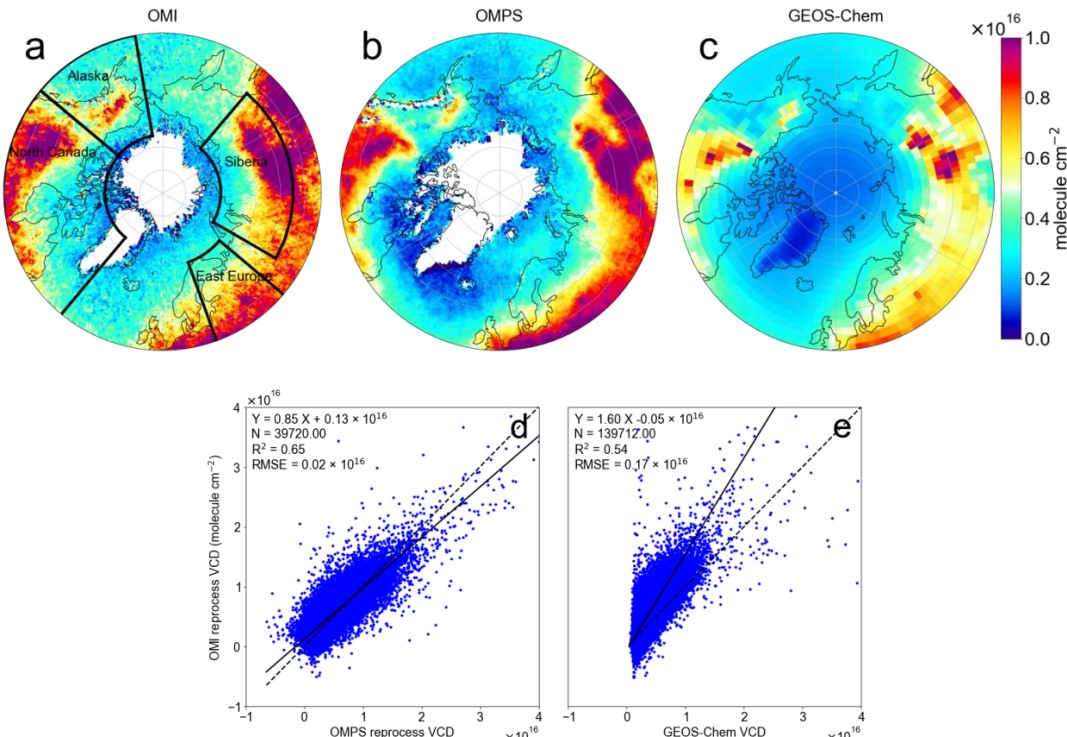

*Figure 1 | HCHO VCD from OMI, OMPS-SNPP and GEOS-Chem, as well as their linear*

*correlation. (a), (b) and (c) shows Spatial pattern of July mean HCHO VCD from reprocessed OMI, reprocessed OMPS-SNPP and GEOS-Chem over northern high latitudes, in 2012-2019 summers. The black boxes in (a) show the four study domains: Alaska ([50,75] °N, [-170, -130] °E), Siberia ([57,75] °N, [60,140] °E), North Canada ([50,75] °N, [-130, -40] °E), East Europe*



*([50,71] ˚N, [20,50] ˚E). (d) Scatter plot of monthly HCHO VCD from reprocessed OMI versus*

*reprocessed OMPS-SNPP over continental northern high latitudes in 2012-2019 summers. OMI*

*and OMPS-SNPP data are regridded to 2° × 2.5° horizontal resolution to matchup with GEOS-*

*Chem pixels. (e) is similar to (d) but shows OMI versus GEOS-Chem.*

We examine the HCHO VCD along with biogenic and wildfire emissions over Alaska, Siberia,

North Canada and East Europe from 2005 to 2019. As shown in Figure 2, both satellites and model

show similar intra-annual and interannual variability of summertime HCHO VCD over four

domains. The interannual variability from both model and satellite, with high HCHO VCD in July,

are mainly driven by seasonality of surface temperature, related emissions and chemical

production of HCHO. Alaska appears to have the weakest seasonal variation amongst four

domains, suggesting a small contribution of biogenic emission in HCHO interannual variability.

Consistent with Figure 1, both OMPS-SNPP and OMI HCHO VCDs are higher than modelled

values, with largest discrepancies in July. We find from Figure 2 that high HCHO years are often

associated with strong wildfire emissions in Alaska, North Canada and Siberia, and to a lesser

extent associated with biogenic emissions; while in East Europe, high HCHO years are associated

with large biogenic emissions. This indicates different drivers of interannual variabilities of HCHO

VCD among these four regions.



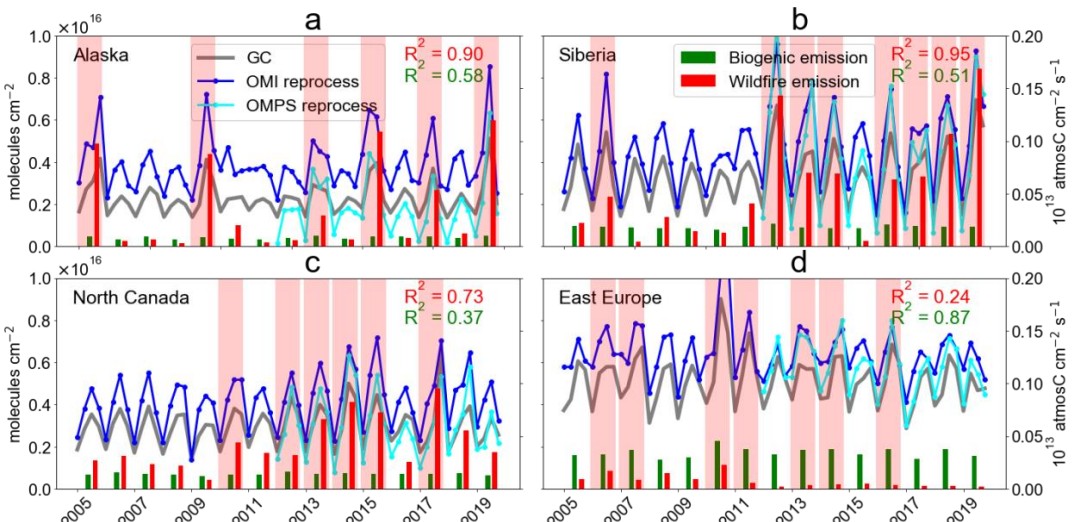

*Figure 2 | Timeseries of HCHO VCD, biogenic and wildfire emissions over (a) Alaska, (b) Siberia, (c) North Canada and (d) East Europe, May 1-August 31, 2005-2019. The blue lines are monthly HCHO VCD from reprocessed OMI, cyan lines are from reprocessed OMPS-SNPP, grey lines are from GEOS-Chem. Red bars are area-normalized wildfire carbon emissions during the summer of each year; green bars are area-normalized biogenic VOC carbon emissions. Pink shade indicates high HCHO VCD years (definition see Section 2.2 and Table S1). The $R^2$ between GEOS-Chem HCHO VCD and biogenic emission (green) / wildfire emission (red) is shown at top right of each panel.*

We now use CV to quantify the interannual variability of HCHO VCD from model and satellites over the four regions. We show from Figure 3 that OMI and OMPS-SNPP have a similar interannual variability of HCHO VCD over each region, with CV ranging from ~10% to ~15%. Both OMI and OMPS-SNPP show highest interannual variability of summertime HCHO VCD over Alaska, and lowest interannual variability over East Europe. GEOS-Chem suggests a similar but lower CV, ranging from 5% to 10%. We further examine the contribution from background,



biogenic and pyrogenic emissions to the CV over each region. We find from model results that biogenic emission and background signal contributes to 90% of the interannual variability of

HCHO VCD in East Europe, while wildfire accounts for over 90% of CV in Alaska, Siberia and North Canada, consistent with previous work (Stavrakou et al., 2018; Zhao et al., 2022). Using Mann-Kendall test, we found no significant trend of HCHO VCD over East Europe and Alaska from either satellites or model. On the other hand, we find the trend over Siberia and North Canada is significant ($p<0.05$) and increasing. $VCD_{0,GC}$ and $dVCD_{Bio,GC}$ show no significant trend, while

the trend of $dVCD_{Fire,GC}$ is significant and increasing in Siberia and North Canada, suggesting that wildfires are responsible for the $VCD_{GC}$ trends in these two regions. In contrast to Bauwens et al (2016), we find that HCHO VCD trend over Siberia and North Canada is largely driven by the increasing wildfires in recent years, and to a lesser extent by biogenic VOC emissions.

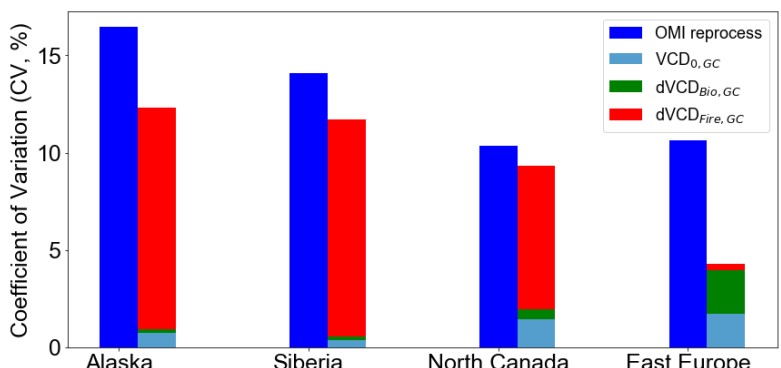


*Figure 3 | Coefficient of Variation (CV) of OMI HCHO VCD and modelled HCHO dVCDs in*

*summertime of 2005-2019.*



### 4. Main drivers of HCHO VCD interannual variabilities

Figure 4 shows a similar spatial pattern of $VCD_{0,GC}$ and $dVCD_{Bio,GC}$, with a distinctive spatial

pattern of $dVCD_{Fire,GC}$. $VCD_{0,GC}$ accounts for 66-87% of HCHO VCD in low HCHO years and

~50-62% in high HCHO years. $dVCD_{Fire,GC}$ shows significant enhancements over Central Siberia

and North America, with values larger than $5\times10^{15}$ molecules $cm^{-2}$ at fire hot spots. $dVCD_{Bio,GC}$

spatial pattern corresponds mainly to isoprene emissions over vegetated area, with high values

over East Europe ($3\times10^{15}$ molecules $cm^{-2}$) and East Siberia ($4\times10^{15}$ molecules $cm^{-2}$). Model

suggests that East Europe is covered by needle leaf evergreen temperate trees and broadleaf

deciduous boreal trees, while East Siberia is mainly covered by needle leaf evergreen boreal trees

(Figure S1). We note that $\Delta dVCD_{Bio,GC}:\Delta ISOPe$ (Isoprene emission flux. Unit: $10^{16}$ molecules $cm^{-2}$

per $10^{13}$ atmosC $cm^{-2}$ $s^{-1}$) over northern high latitudes is around 0.24, a factor of 10 lower than

$\Delta VCD_{GC}:\Delta ISOPe$ over Southeast US (Millet et al., 2008). This indicates a much lower HCHO

production efficiency from isoprene oxidation in northern high latitudes compared to mid-latitude,

possibly resulting from the availability of $NO_x$ (Marais et al., 2012; Mao et al., 2013; Wolfe et al.,

2016).

Our modelled ISOPe is ~1-2 times higher than MONOe (monoterpenes emission) in Alaska,

Europe, North Canada and central Siberia boreal forest zone, as shown in Figure 4(d) and (e). Our

model can largely reproduce isoprene surface mixing ratios along Trans-Siberian Railway within

Russian boreal forests (generally <1ppb in our model, and around 0.31–0.48 ppb in the in-situ

campaign in Timkovskys et al (2010), both can reach ~4 ppb in East Siberia). Our model also

reproduces monoterpenes surface mixing ratios over Alaskan North Slope (0.009 ppbv in our

model and ~0.014 ppbv in Selimovic et al (2022)). Comparing to Stavrakou et al (2018), our



modeled ISOPe over East Europe, Alaska and North Canada agrees within 20%, but our modeled

MONOe is around 40% lower, likely due to different PFT map and meteorological fields

(Guenther et al., 2012).

A remarkable feature is the heterogeneity of BVOC emissions in northern high latitudes revealed

by measurements. We show in Table 2 that while isoprene dominates BVOC emission over the

Arctic tundra and broadleaf forests, monoterpene becomes the dominated species over coniferous

forests. This includes a large portion over European boreal zone, such as at Hyytiälä in Finland

(Rinne et al., 2000; Bäck et al., 2012; Rantala et al., 2015; Zhou et al., 2017), Bílý Kříž in Czech

Republic (Juráň et al., 2017) and Norunda research station in Sweden (Wang et al., 2017). However,

this large-scale heterogeneity is not being reproduced by our model. We find from Figure 4(f) that

modeled BVOC emissions are dominated by isoprene in most part of northern high latitudes,

except East Siberia and East Greenland. As shown in Figure S1, the isoprene-dominated region is

mainly due to broad-leaf deciduous boreal shrubs and needle-leaf evergreen boreal trees that are

assumed in the model and exhibits higher isoprene emission factors than monoterpenes; in contrast,

East Siberia is covered predominantly by needle-leaf deciduous boreal trees, leading to higher

monoterpenes than isoprene emission (Guenther et al., 2012). We expect the discrepancy between

model and measurement to have small effects on HCHO VCD, as BVOC generally contributes to

less than 9-37% of HCHO VCD in high HCHO years and 12-27% in low HCHO years.




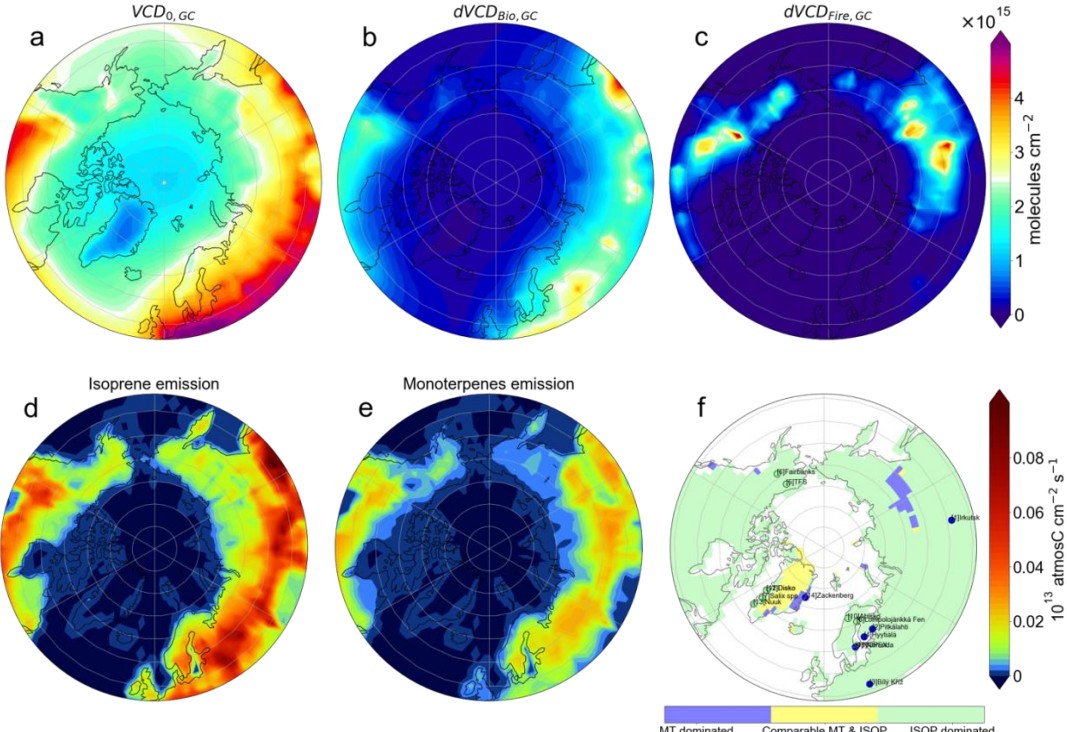

*Figure 4| (a) to (e) GEOS-Chem HCHO $VCD_{0,GC}$, $dVCD_{Bio,GC}$, $dVCD_{Fire,GC}$, isoprene and monoterpenes emission fluxes over northern high latitudes, averaged for July from 2005 to 2019.*

*(f) BVOC emission regimes over northern high latitudes, in GEOS-Chem simulation for 2005-2019 summers and from in-situ measurements (references listed in Table 2). Isoprene-dominates regime at a pixel means isoprene emission is significantly higher than monoterpenes emission (p<0.05 in t-test) for May-August in 2005-2019.*


**Table 2. In-situ measurements of BVOC in Figure 4(f)**

| Site name | Lat(°N), Lon(°E) | Period of the measurement | Major vegetation type | Predominant BVOC | References |
|-----------|------------------|---------------------------|------------------------|------------------|------------|





| | | | at location of measurement | | |
|---|---|---|---|---|---|
| [1] Irkutsk | 53.00, 102.27 | July 21–August 4, 2008 | Boreal coniferous forest | Monoterpenes | Timkovsky et al., 2010 |
| [2] Pitkälahti | 62.72, 30.96 | May to October, 1997-1998 | mixed forest | Monoterpenes | Hakola et al., 2000 |
| [3] Bílý Kříž | 49.50, 18.54 | Summer, 2009-2014 | Norway spruce forest | Monoterpenes | Juráň et al., 2017 |
| [4] Hyytiälä | 61.84, 24.29 | 1) August, 2001 2) May, 2010 to December, 2013 3) April to November 2008 | Boreal coniferous forest (Scots pine (Pinus sylvestris) and Norway spruce (Picea abies)) | Monoterpenes | Spirig et al., 2004, Rantala et al., 2015, Aaltonen et al., 2011 |
| [5] TFS | 68.63, -149.59 | May–June 2019 | Arctic Tundra | Isoprene | Angot et al., 2020; Selimovic et al., 2022 |
| [6] Fairbanks | 64.84, -147.72 | August 2016 | Needle-leaf evergreen boreal forest | Isoprene | Zhao et al., 2022 |
| [7] Kangerlussuaq (Salix spp.) | 67.01, -50.73 | late June to early August 2013 | Salix spp. | Isoprene | Vedel-Petersen et al., 2015 |
| [8] Lompolojänkkä | 66.61, 24.06 | May to August, 2018 | Sub-Arctic fen | Isoprene | Hellén et al., 2020 |
| [9] NOPEX site | 60.08, 17.50 | May to September, 1995 | mixed forest | Isoprene | Janson et al., 1999 |
| [10] Abisko Scientific Research Station | 68.35, 18.82 | June to August, 2006-2007,2010-2012 | Subarctic wet heath | Isoprene | Tiiva et al., 2008; Faubert et al., 2010; Valolahti et al., 2015 |



| [11] Disko | 69.24, -53.53 | June to August, 2013-2014 | Subarctic heath | Monoterpenes | (Lindwall et al., 2016a) |
|---|---|---|---|---|---|
| [12] Disko | 69.24, -53.53 | June to August, 2014-2015 | Arctic fen | Isoprene | (Lindwall et al., 2016b) |
| [13] Nuuk | 64.12, -51.35 | June to August, 2013 | Subarctic heath | Isoprene | Kramshøj et al., 2016 |
| [14] Zackenberg | 74.50, -20.50 | August, 2009 | Mesic to dry mixed heath | Monoterpenes | (Schollert et al., 2014) |
| [15] Norunda | 60.08, 17.48 | June to September, 2013 | Norway Spruce | Monoterpenes | (Wang et al., 2017) |

Figure 5(a) to (c) shows that wildfire is the main driver of HCHO VCD interannual variability over Siberia, north Canada and Alaska. In low HCHO years of these three domains, $dVCD_{Fire,GC}$

contribution ~5-10% of HCHO total column, less than $VCD_{0,GC}$ and $dVCD_{Bio,GC}$; in high HCHO years, $dVCD_{Fire,GC}$ contribution to total column rises to ~20-40%. This is consistent with Figure 2 that HCHO VCD interannual variability have significantly higher correlations with wildfire emissions than with biogenic emission over Siberia, north Canada and Alaska. These findings highlight the role of wildfire in driving HCHO interannual variability in the three domains.




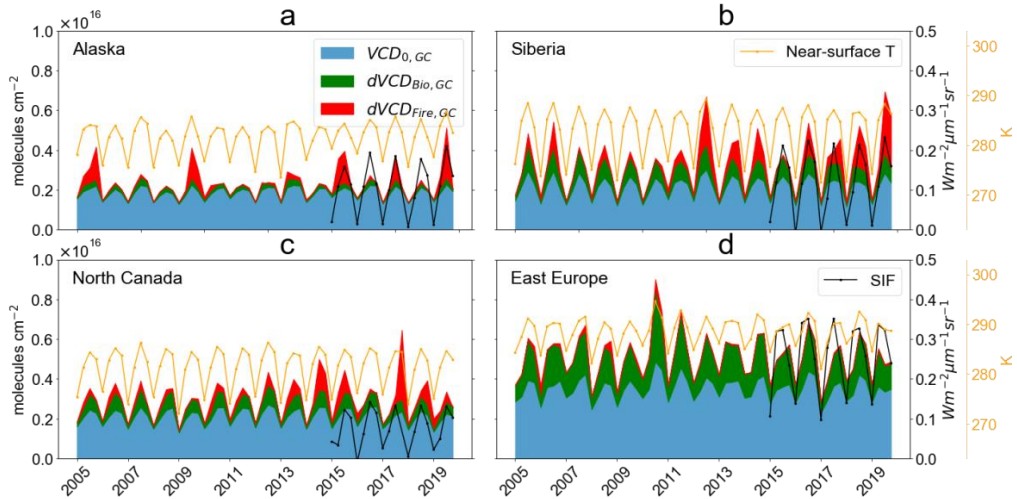

*Figure 5 | Interannual variability of monthly HCHO $VCD_{0,GC}$, $dVCD_{Bio,GC}$ and $dVCD_{Fire,GC}$ as well*

*as near-surface temperature over (a) Alaska, (b) Siberia, (c) North Canada and (d)East Europe,*

*in 2005-2019 summers. The indigo, green and red shades are background HCHO $VCD_{0,GC}$,*

*$dVCD_{Bio,GC}$ and $dVCD_{Fire,GC}$, based on GEOS-Chem sensitivity tests (Table 1). The orange curves*

*are monthly surface temperature from MERRA-2 dataset. The black curves are OCO-2 monthly*

*SIF.*

In East Europe, biogenic emission and background HCHO accounts for the majority of HCHO

VCD interannual variability, largely due to surface temperature and availability of $NO_x$. In Figure

5(d), the surface temperature in East Europe is higher than that in Alaska, North Canada and

Siberia by 5-7 K, leading to an increase in BVOC emissions and $VCD_{0,GC}$ through methane

oxidation. HCHO VCD is further enhanced through the higher $NO_x$ level (0.4-1ppbv) in East

Europe than in other three domains (0.1-0.5ppbv), as HCHO yield from isoprene photooxidation

increases with $NO_x$ level. The high $NO_x$ level in East Europe results from its large urban areas and



high anthropogenic emissions. The large contribution of BVOC to HCHO VCD is consistent with Figure 3, which shows the CV of $dVCD_{Bio,GC}+VCD_{0,GC}$ accounts >90% of $VCD_{GC}$'s CV in East Europe. Similarly, Figure 2(d) shows that biogenic emission has a higher correlation ($R^2 = 0.87$) with $VCD_{GC}$ than wildfire emission does ($R^2=0.24$). These results suggest that biogenic emission and background are the main contributors of HCHO interannual variability in East Europe.

### 5. SIF evaluation on $dVCD_{Bio,GC}$ interannual variability

To further evaluate the drivers of interannual variability of HCHO VCD, we examine the correlation between SIF and HCHO signal from various regions. In Figure 6, SIF and $dVCD_{Bio,GC}$ or ISOPe show better coupling under a lower SIF level, possibly due to the different temperature optimums of isoprene emission and photosynthesis (Harrison et al., 2013; Zheng et al., 2015). We calculated the correlation via Standardized Major Axis (SMA) regression for SIF within 0-0.25 $Wm^{-2}\mu m^{-1}sr^{-1}$. Figure 6(a)-(d) show a similar linear regression slope between SIF and $dVCD_{Bio,GC}$ over the East Europe, Siberia and North Canada, a factor of 3-4 higher than the slope over Alaska. The good correlation between SIF and $dVCD_{Bio,GC}$ is expected, as both are largely driven by surface temperature (Figure S2 and S3) . Despite the difference in distribution of vegetation types, the similar $dVCD_{Bio,GC}$-SIF slopes over Siberia, North Canada and East Europe (slope=0.27-0.43, unit:$10^{16}$ molecules $cm^{-2}$ per $Wm^{-2}\mu m^{-1}sr^{-1}$), indicates SIF as a proxy of $dVCD_{Bio,GC}$ spatiotemporal variability in these domains. The low $dVCD_{Bio,GC}$-SIF slope in Alaska warrants further investigation.

We further examine the relationship between ISOPe and SIF. We find ISOPe:SIF slopes to be less uniform compared to $dVCD_{Bio,GC}$-SIF slopes, likely due to the widespread enhancement of HCHO





VCD that largely reduces the spatial gradient of isoprene emissions (Zhao et al. 2022). In contrast

to high latitudes, we find that both ISOPe:SIF slope and dVCD$_{Bio,GC}$:SIF slope are significantly

higher in Southeast US and Amazon (Figure 6(e)-(f), (k)-(l)), suggesting much stronger isoprene

emissions per unit of SIF at lower latitudes. Foster et al (2014) show a high linear correlation

between seasonal variation of satellite HCHO column (fire free) and GPP in northern high latitudes.

This is consistent to our finding that dVCD$_{Bio,GC}$ and SIF are highly correlated in northern high

latitudes (Figure S2), since SIF is a widely used proxy of GPP (Frankenberg et al., 2011).

SIF offers an independent evaluation on the interannual variability of BVOC emissions. As SIF

are tightly correlated with dVCD$_{Bio,GC}$ and isoprene emissions, it is reasonable to infer from Figure

5 that the low interannual variability shown in SIF is expected for dVCD$_{Bio,GC}$ and isoprene

emissions in four domains. In contrast, we find a much weaker correlation between SIF and

dVCD$_{Fire,GC}$, suggesting that the low interannual variability of SIF cannot be applied to wildfire

emissions and their contribution to HCHO VCD.  As wildfire emission is highly correlated with

HCHO interannual variability over North Canada, Siberia, and Alaska (Figure 2), it is unlikely

that the strong HCHO interannual variabilities are driven by biogenic emissions.

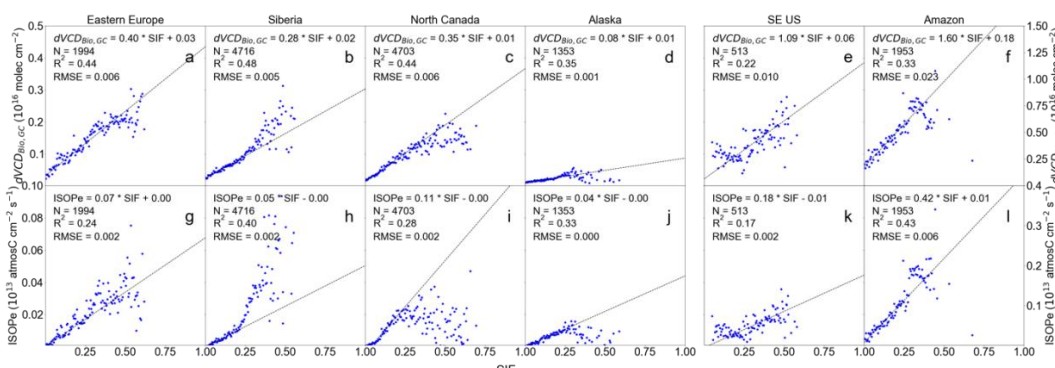

440



*Figure 6 | Scatter plot of monthly OCO-2 SIF versus GEOS-Chem HCHO dVCD$_{Bio,GC}$ and isoprene emission fluxes in the four study domains plus Southeast US ([26, 36]°N, [-100,-75]°E) and Amazon([-20,-5]°N, [-75,-40]°E), from May to August in 2015-2019. OCO-2 SIF is regridded to 2° × 2.5° spatial resolution. Only continental pixels of SIF-dVCD$_{Bio,GC}$ and SIF-ISOPe matchups are used to plot. Before plotting, data matchups are binned by SIF, using a bin size of 0.005 Wm$^{-2}$μm$^{-1}$sr$^{-1}$. SMA regression is shown as the black dash in each panel, calculated for SIF within 0-0.25 Wm$^{-2}$μm$^{-1}$sr$^{-1}$.*

## 6.  Conclusions and discussions

We use reprocessed new retrievals of HCHO from OMI and OMPS-SNPP to evaluate the interannual variability of HCHO VCD from GEOS-Chem over northern high latitudes in 2005-2019 summers. The reprocessed OMI and OMPS-SNPP HCHO VCDs show a high consistency in the spatial pattern and interannual variability. GEOS-Chem reproduced the interannual variability of HCHO VCD but the magnitude is biased low comparing to satellite retrievals.

Wildfire accounts for the majority of HCHO interannual variability in Alaska, North Canada and Siberia. Compared to biogenic emissions and background HCHO, wildfire emission shows a better correlation with HCHO VCD, despite that biogenic and background HCHO can dominate HCHO VCD in low HCHO years of these three regions. We also find an increasing trend ($p < 0.05$) in wildfire emission and HCHO VCD over North Canada and Siberia. In fact, our modeled HCHO VCD can be biased low, due to large underestimate of HCHO from wildfire emissions (Liao et al., 2021; Liu et al., 2017; Permar et al., 2021). With rapid Arctic warming, wildfire frequency and

intensity rises rapidly in recent decades and near future (Descals et al., 2022). We expect wildfire
continues to dominate HCHO interannual variability in the three regions.


East Europe is the only one of the four studied regions where HCHO interannual variability is
dominated by biogenic emission and background HCHO. This is due to a combination of lower
wildfire activities, higher surface temperature and anthropogenic $NO_x$ emissions in this region. No
significant trend of biogenic emission, biogenic-related HCHO and background HCHO are found

in the four regions during summertime of 2005-2019. However, model estimate of HCHO from
biogenic emissions are largely uncertain, as model calculated VOC speciation is at odds with field
measurements (Figure 4(f) and Table 2). Previous work shows good performance of model in
capturing long-term variability of biogenic emission in response to climate variables (Stavrakou
et al., 2018), but model underestimates biogenic and fire emissions over northern high latitudes,

especially over East Europe and Alaska (Stavrakou et al., 2015). Future research is warranted to
examine the HCHO signal from biogenic emissions in this region.

The OCO-2 satellite SIF provides an additional constraint on the interannual variability of biogenic
emissions and is independent of wildfire emissions. As a proxy of vegetation photosynthesis and

GPP, SIF is expected to have a good correlation with isoprene emission and HCHO VCD in the
northern boreal regions, though this correlation can be worse in mid-latitudes and tropical region
(Foster et al., 2014). We show a tight correlation between SIF and $dVCD_{Bio,GC}$ , and between SIF
and isoprene emissions at northern high latitudes, suggesting that SIF can be used as a proxy for
isoprene emissions in this region. It remains unclear why the $dVCD_{Bio,GC}$-SIF slope in Alaska is



lower than other domains. SIF may serve as a tool to understand biogenic emissions at northern

high latitudes.

**Code and data availability.**

The        OMPS-SNPP        HCHO        L2        product        is        available        at

https://disc.gsfc.nasa.gov/datasets/OMPS_NPP_NMHCHO_L2_1/summary    (González Abad,

2022). The OMI HCHO L2 product is available at (xxxxx). The OCO-2 SIF is available at

https://daac.ornl.gov/cgi-bin/dsviewer.pl?ds_id=1863 (Yu et al., 2021). Data used in this work is

available at https://doi.org/10.6084/m9.figshare.23599566.v1 (Zhao, 2023a). Data processing and

plotting codes are available at https://doi.org/10.5281/zenodo.8094844 (Zhao, 2023b). The GEOS-

Chem model is publicly available at: https://doi.org/10.5281/zenodo.3701669 (GEOS-Chem,

2020).

**Supplement.**

The supplement related to this article is available online at: (xxxxx).


**Author contributions.**

TZ and JM designed the research, performed the simulations and conducted the analysis. ZA, GGA

and CN provided OMI and OMPS data. YZ helped process and analyze the data. TZ and JM wrote

the paper with all co-authors providing input.


**Competing interests.**

The contact author has declared that neither they nor their co-authors have any competing interests.



**Disclaimer.**

Publisher's note: Copernicus Publications remains neutral with regard to jurisdictional claims in

published maps and institutional affiliations.

**Acknowledgement.**

TZ and JM acknowledge funding from NASA grant 80NSSC19M0154 and 80NSSC21K0428.

ZA, GGA and CN acknowledge funding from NOAA grant NA18OAR4310108 and NASA

grants 80NSSC18M0091, 80NSSC18K0691 and 80NSSC21K0177. We thank William Simpson

(University of Alaska Fairbanks) for helpful discussions.

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
