# Peer review of "Interannual variability of summertime formaldehyde (HCHO) vertical column density and its main drivers in northern high latitudes"

_EGUsphere, 2023_

## Referee Comment (RC2)

**Overall Feedback:**

Zhao and co-authors investigated the summertime HCHO interannual variability in northern high latitudes. Using the GEOS-Chem model and satellite data, they highlighted that wildfire is the major driver in regions like Siberia, Alaska, and North Canada. Yet, biogenic emissions and methane oxidation predominantly drive HCHO variability in East Europe. They also introduce SIF as a potential indicator for biogenic emissions in these northern regions. However, there are specific concerns with the current manuscript that need to be addressed before it can be considered for acceptance.

**These concerns are detailed below.**

**Major scientific concerns:**

1. **Linear Relationship Assumption:**

Formaldehyde's dynamics are significantly influenced by secondary productions. Given this, is it appropriate to assume a linear relationship across various sources? How have you addressed the discord between the nonlinear chemistry and your linear assumptions? Moreover, how might this nonlinearity affect your conclusions?

2. **Scope of SIF Analysis:**

The SIF analysis is restricted to a range between 0 to 0.25. While the slope derived might be relevant for higher SIF values in lower latitudes (Figure 6), three of four domains in Figure 5 pertains to higher latitudes. This raises questions on the applicability of the derived relationship to VCD and emission data within these domains. How do you reconcile this?

3. **MEGAN Computation in GEOS-Chem:**

While MEGAN is computed online within GEOS-Chem, there is a significant divergence between online and offline MEGAN results, as noted in the discussion below. Since GEOS-Chem typically recommends having both online and offline modes enabled by default, could the exclusive use of online MEGAN have swayed your results?

Link: http://wiki.seas.harvard.edu/geos-chem/index.php/MEGAN_v2.1_plus_Guenther_2012_biogenic_emissions#Note_on_the_use_of_pre-computed_emission_factor_.28EF.29_maps_versus_EF_maps_computed_online

4. **Potential Biogenic Underestimation of GEOS-Chem:**

Selimovic et al. (2022) showed that GEOS-Chem underestimates OVOCs including methanol, formaldehyde, formic acid and acetic acid by a factor of 3 to 12 in arctic area. Among them, the negative model bias for methanol is attributed to outdated MEGAN. In light of this, is it conceivable that your simulated biogenic emissions are similarly underestimated? What implications could this have on your study's conclusions?

*Reference*: Selimovic, Vanessa, et al. "Atmospheric biogenic volatile organic compounds in the Alaskan Arctic tundra: constraints from measurements at Toolik Field Station." Atmospheric

Chemistry and Physics 22.21 (2022): 14037-14058.
https://acp.copernicus.org/articles/22/14037/2022/acp-22-14037-2022.pdf

**General writing issues:**

**1. Adherence to Writing Guidances:**

Please ensure that your manuscript aligns with the ACP writing guidelines stipulated. For example, when referring to a section in the running text, utilize the abbreviation "Sect." followed by the respective number, unless the reference initiates a sentence. Thoroughly review the ACP writing guidance to ascertain that the manuscript conforms to all provided specifications.

**2. Quantitative Precision:**

There are too many vague words used in the paper. Please aim to bolster the precision and clarity of your assertions by providing quantitative data wherever feasible. Avoid ambiguous or generalized statements; instead, substantiate your claims with numbers. Not only for your abstract but the main texts as well.

**3. Citation Authenticity:**

It's paramount that you intimately familiarize yourself with the papers you cite. This ensures the citations support your arguments. Refrain from randomly citing papers merely to bolster the appearance of your statements. Authentic and relevant citations strengthen the validity of your work and demonstrate thorough research diligence. While I have not looked into every reference in your manuscript, I have identified some citations that appear to be misaligned or inappropriate.

**4. Conciseness in Writing:**

Many sections of your manuscript contain superfluous or verbose language. Aim for succinctness, ensuring that each word adds value to your statements. Make every word count.

**Other comments for each section:**

**Abstract:** Please write in a quantitative way.

**Introduction:**

Line 29-30: Consider providing a foundational and more general reference to support this statement, rather than solely referencing works from your co-authors.

Line 43: Avoid just using words such as "large". Provide numbers.

Line 49: The word "predominant" is confusing. What do you mean? The most abundant? The most reactive? or both?

Line 53: Same for line 49.

Line 53-59: The connection or relevance of these points is unclear. Could you clarify the relationship?

Line 60: Given the focus on formaldehyde in the preceding and following lines, I guess you want to say the "Wildfire is another important source of formaldehyde"? If so, ensure each word has impact.

Line 63: It seems the biogenic and the background are still the main CH2O sources but fire may be the driven for interannual trend. Please double check your statements here.

Line 70: Why using SIF? Is that because other methods having limits? Please provide more context here.

Line 80-81: Do readers also need to know pros and cons of OMI and OMPS before understanding your work?

**Observations and Model:**

Sect. 2.1: Again, why using OMI and OMPS? What are their pros and cons? How does use these two datasets help each other?

Sect. 2.2:

Line 164-173: Have you done the spin up? Also, please use right reference of GEOS-Chem model. For example, in line 176, I believe van der Werf et al. (2017) is the one for GFED4s. Also you need to cite the GEOS-Chem doi with corresponding version. Please read through the web below in GC website.

https://geos-chem.seas.harvard.edu/geos-chem-narrative

http://wiki.seas.harvard.edu/geos-chem/index.php/GEOS-Chem_versions

Line 178-179: Please cite the original reference of 3-hourly scale factor. You need to read the paper and give credits to people who contribute to the work.

Line 181-182: I believe either Liu or Permar was working on the western US, which is mostly temperate forest. I do not think they are appropriate to compare to boreal forest. If you want to compare boreal forest, you need to cite paper for EF reports from boreal forest or at least mention the EF reference for GFED4. Again, you need to give credits to people who contribute to the work.

Line 185: Results from online and offline MEGAN computation is significantly different. See the web below. I believe offline and online are both turned on by default, which is recommended by GEOS-Chem team. Would the solely online MEGAN influence your results? How do you resolve it?
http://wiki.seas.harvard.edu/geos-chem/index.php/MEGAN_v2.1_plus_Guenther_2012_biogenic_emissions#Note_on_the_use_of_pre-computed_emission_factor_.28EF.29_maps_versus_EF_maps_computed_online

Line 193: Did you explicitly represent the chemistry for isomers or just use the default lumping chemistry for monoterpenes? If I understand it well, the monoterpenes you used is lumped monoterpenes (no?).

Line 195-205: Read through narrative description for more information (web provided above). Using MEGAN as an example, you need to cite Hu et al. (2015) as well. Example of how to cite MEGAN in GEOS-Chem narrative description: Biogenic VOC emissions in GEOS-Chem are from the MEGAN v2.1 inventory of Guenther et al. [2012] as implemented by Hu et al. [2015b].

Line 197: I believe the isoprene chemistry is updated by Kelvin Bate. Again, please check the narrative description!

Line 210: Say three simulations a series of GEOS-Chem simulations. Be specific of what you are trying to write.

Line 211: Change ":" into "including" might be better?

Line 215: Is it fair to assume the equation is linear for VOCs can be highly impacted by secondary production? How do you resolve the nonlinearity here and how it would influence your conclusion?

Line 239-240: Again, is the assumption fair? See above comments for line 215 instead.

**OMI/OMPS Evaluation with GEOS-Chem HCHO VCD**

Line 250-253: I did not fully read Nowlan et al. (2023) but are you suggesting the correction in that paper does not fix the "high-latitudes bias"?

Line 255-258: What's the satellite retrievals uncertainty here? Try to be specific here.

Line 258-259: What is the "biases in both model and satellites"? Try to be specific rather than using vague words.

Figure 2: Is the carbon emissions from emission inventories?

Line 312: What is Mann-Kendall test?

Line 315-320: What cause the inconsistency here? Any implications we can gain here?

**Main drivers of HCHO VCD interannual variabilities**

Line 330: Again, does the vegetation production matter? See my comments for MEGAN.

Line 334-335: There are other potential reasons. NOx level is one for sure. It could also relate to the difference of temperature, photolysis, and oxidants level.

Line 340-345: Might need to weaken the words? You are comparing the campaign-average with your custom model results rather than directly compare to the observation at the exactly time and location if I understand it right.

Line 347: It should be easy to use the recommended veg map in the GEOS-Chem and then figure out if it is the reason (no?).

Figure 5: Is that seasonally or monthly HCHO? It seems that you only have four data points for each year.

Line 395: See comments for Line 334-335

Line 397: I might not be familiar with such work but how are BVOC and methane oxidation sensitive to varying temperature?

**SIF evaluation on dVCDBio,GC interannual variability**

Line 412-413: What is the specific reason for choosing the Standardized Major Axis (SMA) regression?

Line 415-430: As both SIF and dVCDBio,GC are largely driven by surface temperature, why not just using surface temperature instead? What's the implication of SIF ranges (0-0.25)? What's the benefit of using SIF rather than other proxy?

**Conclusions and discussions**

Line 462: Jin2021 highlighted the importance of OH in HCHO production. Liu and Permar reported the observed fire ER/EF. How do they suggest underestimated HCHO in wildfire emission?

---

## Author Comment (AC1)

Response to Reviewer #2

We are grateful to the reviewer for the valuable comments that facilitate the important improvements of the original manuscript. We list the point-by-point responses below. The reviewer's comments are marked black and our responses are marked dark blue. Line numbers refer to the discussion paper egusphere-2023-1431.

Zhao and co-authors investigated the summertime HCHO interannual variability in northern high latitudes. Using the GEOS-Chem model and satellite data, they highlighted that wildfire is the major driver in regions like Siberia, Alaska, and North Canada. Yet, biogenic emissions and methane oxidation predominantly drive HCHO variability in East Europe. They also introduce SIF as a potential indicator for biogenic emissions in these northern regions. However, there are specific concerns with the current manuscript that need to be addressed before it can be considered for acceptance.

Major scientific concerns:

1. Linear Relationship Assumption: Formaldehyde's dynamics are significantly influenced by secondary productions. Given this, is it appropriate to assume a linear relationship across various sources? How have you addressed the discord between the nonlinear chemistry and your linear assumptions? Moreover, how might this nonlinearity affect your conclusions?

We performed model sensitivity tests to examine the impact of non-linear effect on HCHO VCD constitution assumption. We modify the description in Sect. 2.2: "

To examine the influence of different sources on HCHO columns in northern high latitudes, we conducted four GEOS-Chem simulations, as described in Table 1, to separate modelled HCHO total column ($VCD_{GC}$) into three parts, including the background column ($VCD_{0,GC}$), biogenic emission induced column ($dVCD_{Bio,GC}$) as well as wildfire emission induced column ($dVCD_{Fire,GC}$):

$$VCD_{GC} = VCD_{0,GC} + dVCD_{Bio,GC} + dVCD_{Fire,GC} \quad (3)$$

$VCD_{0,GC}$ is the $VCD_{GC}$ from the GEOS-Chem simulation in which both biogenic and wildfire emissions are turned off. $VCD_{0,GC}$, $dVCD_{Fire,GC}$ and $dVCD_{Bio,GC}$ are derived by Eq. (4a) to (4c):

$$VCD_{0,GC} = VCD_{GC}(BG) \quad (4a)$$
$$dVCD_{Fire,GC} = VCD_{GC}(All) - VCD_{GC}(NF) \quad (4b)$$
$$dVCD_{Bio,GC} = VCD_{GC}(NF) - VCD_{0,GC} \quad (4c)$$
$$dVCD_{Fire,GC}* = VCD_{GC}(NB) - VCD_{GC}(BG) \quad (4d)$$
$$dVCD_{Bio,GC}* = VCD_{GC}(All) - VCD_{0,GC}(NB) \quad (4e)$$

To assess the linearity assumption in Eq. (3), we conducted model sensitivity tests over a one-month period to evaluate the disparity between $VCD_{GC}$ and $VCD_{0,GC} + dVCD_{Fire,GC}* + dVCD_{Bio,GC}*$ (derived from Eq.(4a), (4d) and (4e)). The difference between these two terms is

less than 14% in northern high latitudes, suggesting a minor importance of the non-linear effect in this area.

**Table 1.** Configuration of GEOS-Chem global simulations in this study

| Simulations | Biogenic emission | Wildfire |
|---|---|---|
| Biogenic + wildfire + Background (All) | On | On |
| Background (BG) | Off | Off |
| Biogenic + Background (NF) | On | Off |
| Wildfire + Background (NB) | Off | On |

"

2. Scope of SIF Analysis: The SIF analysis is restricted to a range between 0 to 0.25. While the slope derived might be relevant for higher SIF values in lower latitudes (Figure 6), three of four domains in Figure 5 pertains to higher latitudes. This raises questions on the applicability of the derived relationship to VCD and emission data within these domains. How do you reconcile this?

We only use the 0 to 0.25 range for SIF to calculate the linear regression lines in Figure 6. In Figure 6, the data amount (N), $R^2$, RMSE are calculated based on all data, not just the data pairs that have SIF within 0 to 0.25. In Figure 5, the regional monthly SIF is also calculated based on all ranges.

We reorganize the paper and will discuss the coupling between SIF and biogenic HCHO in Sect.5.

We add one sentence to the caption of Figure 6 : "

[Figure]

*Figure 6.* *Scatter plot of monthly OCO-2 SIF versus GEOS-Chem HCHO $dVCD_{Bio,GC}$ and isoprene emission fluxes in the four study domains plus Southeast US ([26, 36]°N, [-100,-75]°E) and Amazon([-20,-5]°N, [-75,-40]°E), from May to August in 2015-2019. OCO-2 SIF is regridded to $2°×2.5°$ spatial resolution. Only continental pixels of SIF-$dVCD_{Bio,GC}$ and SIF-ISOPe matchups are used to plot. Before plotting, data matchups are binned by SIF, using a bin*

*size of 0.001 $Wm^{-2}\mu m^{-1}sr^{-1}$. Linear regression is shown as the black dash in each panel,*
*calculated for SIF within 0-0.25 $Wm^{-2}\mu m^{-1}sr^{-1}$.* *Amount of binned data pairs (N), R-Squared ($R^2$),*
*Root Mean Square Error (RMSE) are calculated based on binned data across all ranges.*
"

We modify the description in first two paragraph of Sect. 5 to be "
In Figure 6a to 6d, we find a good linear relationship (R=0.6-0.7) between OCO-2 monthly SIF
and $dVCD_{Bio,GC}$ at Alaska, Siberia, Northern Canada and Eastern Europe. Foster et al (2014)
show a high linear correlation between seasonal variation of satellite HCHO column (fire free)
and GPP in northern high latitudes. This is consistent with our finding over most continental
areas in northern high latitudes (Figure S2), since SIF is a widely used proxy of GPP
(Frankenberg et al., 2011). In Figure 6g to 6j, SIF and ISOPe show a linear relationship when
SIF is within 0-0.25 $Wm^{-2}\mu m^{-1}sr^{-1}$ but tend to decouple when SIF > 0.25 $Wm^{-2}\mu m^{-1}sr^{-1}$, possibly
due to the different temperature optimums of isoprene emission and photosynthesis (Harrison et
al., 2013; Zheng et al., 2015).

Despite the difference in distribution of vegetation types, the $dVCD_{Bio,GC}$-SIF slope is
homogeneous over Siberia, Northern Canada and Eastern Europe (slope=0.28-0.45, unit:$10^{16}$
molecules $cm^{-2}$ per $Wm^{-2}\mu m^{-1}sr^{-1}$), suggesting SIF as a tool to understand biogenic HCHO
variability in these regions. The $dVCD_{Bio,GC}$-SIF slope in Alaska is 3-5 times lower than other
three domains, which warrants further investigation. In contrast to high latitudes, we find that
both ISOPe:SIF slope and $dVCD_{Bio,GC}$:SIF slope are a factor of 2-10 times higher in Southeast
US and Amazon (Figure 6e-6f, 6k-6l) than in northern high latitudes, indicating that the
$dVCD_{Bio,GC}$-SIF slope over northern high latitudes and lower latitudes could be very different.
"

3. MEGAN Computation in GEOS-Chem: While MEGAN is computed online within GEOS-
Chem, there is a significant divergence between online and offline MEGAN results, as noted in
the discussion below. Since GEOSChem typically recommends having both online and offline
modes enabled by default, could the exclusive use of online MEGAN have swayed your results?
Link:
http://wiki.seas.harvard.edu/geoschem/index.php/MEGAN_v2.1_plus_Guenther_2012_biogenic
_emissions#Note_on_the_use_ of_pre-
computed_emission_factor_.28EF.29_maps_versus_EF_maps_computed_online

We explained the reason of using online MEGAN, and add a paragraph to discuss the differences
in BVOC emission and HCHO VCD due to online versus offline MEGAN in Sect. 2.2: "

BVOC emissions in the study are calculated online (Emission factor maps computed online)
using the Model of Emissions of Gases and Aerosols from Nature (MEGAN, v2.1) (Guenther et
al., 2006, 2012) as implemented by Hu et al (2015). Terrestrial vegetation for BVOC emissions
is based on the plant functional type (PFT) distribution derived from Community Land Model
(CLM4) (Lawrence et al., 2011; Oleson et al., 2013). Utilizing online MEGAN simplifies the
investigation of the relationship between BVOC emission patterns and PFTs. CLM4 output
(Figure S1) suggests two major PFTs over northern high latitudes: broadleaf deciduous boreal
shrubs (mainly over the northern and south Alaska, northern Canada and northern Siberia) and

needle leaf evergreen boreal trees (mainly over interior Alaska, northern Canada, south Siberia and the northern part of eastern Europe), both with high emission factors in isoprene and low emission factors in monoterpenes. The southern part of Eastern Europe is dominated by croplands and broadleaf deciduous temperate trees. In this work, 'monoterpenes' from model calculation are lumped monoterpenes, including $\alpha$-pinene, $\beta$-pinene, sabinene and carene.

We conducted a model sensitivity test to assess the difference in BVOC emissions and HCHO $dVCD_{Bio,GC}$ due to online versus offline MEGAN applications. The results of the tests show that the use of online MEGAN has a modest impact on monthly ISOPe and MONOe (25-53% for ISOPe in Alaska, Northern Canada and Eastern Europe, 53% for ISOPe in Siberia; 17-24% for MONOe across the four domains), and provide a similar, isoprene-dominated BVOC emission regime over Alaska, Central Siberia, Northern Canada and Eastern Europe, comparing to results from using offline MEGAN. The difference in $dVCD_{Bio,GC}$ between using online and offline MEGAN is approximately 13-26%, suggesting a minor impact on $dVCD_{Bio,GC}$ and $VCD_{GC}$ variability over northern high latitudes when using online or offline MEGAN.

"

4. Potential Biogenic Underestimation of GEOS-Chem: Selimovic et al. (2022) showed that GEOS-Chem underestimates OVOCs including methanol, formaldehyde, formic acid and acetic acid by a factor of 3 to 12 in arctic area. Among them, the negative model bias for methanol is attributed to outdated MEGAN. In light of this, is it conceivable that your simulated biogenic emissions are similarly underestimated? What implications could this have on your study's conclusions? Reference: Selimovic, Vanessa, et al. "Atmospheric biogenic volatile organic compounds in the Alaskan Arctic tundra: constraints from measurements at Toolik Field Station." Atmospheric Chemistry and Physics 22.21 (2022): 14037-14058. https://acp.copernicus.org/articles/22/14037/2022/acp-22-14037-2022.pdf

We reorganize the paragraphs in Sect. 6 (Conclusions and Discussions), and modify the paragraph that discusses the underestimation in modelled HCHO: "

Our modeled HCHO VCD can be biased low, due to large underestimate of HCHO production and emission factor in wildfire smokes. Previous in-situ campaigns show underestimated emission factors of VOCs in GFED4.1s emission inventory for temperate forests in western US (Liu et al., 2017; Permar et al., 2021), while the bias in VOC emission factor in boreal forest wildfires remains unclear. HCHO underestimation can also be due to the missing HCHO secondary production in wildfire-impacted conditions (Liao et al., 2021; Jin et al., 2023). GEOS-Chem is found to underestimate oxygenated VOCs by a factor of 3 to 12 in some Arctic regions, which could contributes to the bias in modelled HCHO in northern high latitudes (Selimovic et al., 2022). More measurements in Arctic region are needed to reconcile the model-observation discrepancies.
"

We modify the paragraph in Sect. 3 that discusses mode-satellite discrepancy outside wildfire areas: "

The model-satellite discrepancies outside wildfire areas could be also due to model underestimates of oxygenated VOCs (OVOCs), biogenic VOC emissions and biases in satellite HCHO retrieval products. For example, Selimovic et al (2022) found that GEOS-Chem underestimates OVOCs, including HCHO, by a factor of 3-12 at Toolik Field Station in Northern Alaska.
"

General writing issues:

1. Adherence to Writing Guidances: Please ensure that your manuscript aligns with the ACP writing guidelines stipulated. For example, when referring to a section in the running text, utilize the abbreviation "Sect." followed by the respective number, unless the reference initiates a sentence. Thoroughly review the ACP writing guidance to ascertain that the manuscript conforms to all provided specifications.

We change all the "section" to the abbreviation "Sect.".

We change all the 'Figure 1(a)' to 'Figure 1a'.

We change the date format in Table 2, from 'July 21 to August 4, 2008' to 'July 21 to August 4 2008'.

We change all the 'East Europe','North Canada', 'East Siberia' to 'Eastern Europe' and 'Northern Canada' and 'Eastern Siberia'.

2. Quantitative Precision: There are too many vague words used in the paper. Please aim to bolster the precision and clarity of your assertions by providing quantitative data wherever feasible. Avoid ambiguous or generalized statements; instead, substantiate your claims with numbers. Not only for your abstract but the main texts as well.

[revised manuscript text omitted]

"

3. Citation Authenticity: It's paramount that you intimately familiarize yourself with the papers you cite. This ensures the citations support your arguments. Refrain from randomly citing papers merely to bolster the appearance of your statements. Authentic and relevant citations strengthen the validity of your work and demonstrate thorough research diligence. While I have not looked into every reference in your manuscript, I have identified some citations that appear to be misaligned or inappropriate.

We double checked the references and revised the incorrect references, to make sure we cited the proper articles.

We change reference "VOCs are main precursors of tropospheric ozone and secondary organic aerosols, strongly impacting air quality and climate (Atkinson, 2000; Kroll and Seinfeld, 2008; Mao et al., 2018; Zheng et al., 2020). "

In line 176, we change the sentence to "Biomass burning emissions in our simulation are derived from the Global Fire Emission Database (GFED4.1s) inventory (van der Werf et al., 2017; Randerson et al., 2017)."

In line 170-172, we change the URL to GEOS-Chem doi "GEOS-Chem version 12.7.2 is deployed (doi: 10.5281/zenodo.3701669, last access: August 10, 2022) with an update on cloud chemistry (https://github.com/geoschem/geos-chem/issues/906, last access: August 10, 2022). "

In line 178-179, we change the sentence to "Emissions on a 3-hour basis are obtained from MODIS satellite observations, which provide information on fire detection and burning area (Mu et al., 2011; van der Werf et al., 2017)."

In line 181-183, we change the sentence to "The GFED4.1s inventory reports the HCHO emission factor of 1.86 g/kg and 2.09 g/kg dry matter for boreal forest and temperate forest fires (Akagi et al., 2011)."

4. Conciseness in Writing: Many sections of your manuscript contain superfluous or verbose language. Aim for succinctness, ensuring that each word adds value to your statements. Make every word count.

We reorganized the analysis sections and conclusion section, deleted superfluous languages, shown in other revisions.

In Sect. 3, we remove this sentence "We examine the HCHO VCD along with biogenic and wildfire emissions over Alaska, Siberia, Northern Canada and Eastern Europe from 2005 to 2019."

In Sect. 4, we removed this part "Figure 5 shows that OMI and OMPS-SNPP have a similar interannual variability of HCHO VCD over each region, with CV ranging from ~10% to ~15%. Both OMI and OMPS-SNPP show highest interannual variability of summertime HCHO VCD over Alaska, and lowest interannual variability over Eastern Europe. GEOS-Chem suggests a similar but lower CV, ranging from 5% to 10%."

**Other comments for each section:**

**Abstract:** Please write in a quantitative way.

We rewrite abstract like this: "
The northern high latitudes (50-90°N, mostly including boreal forest and tundra ecosystem) has been undergoing rapid climate and ecological changes over recent decades, leading to significant variations in Volatile Organic Compounds (VOCs) emissions from biogenic and biomass burning sources. Formaldehyde (HCHO) is an indicator of VOC emissions, but the interannual variability of HCHO and its main drivers over the region remain unclear. In this study, we use the GEOS-Chem chemical transport model and satellite retrievals from Ozone Monitoring Instrument (OMI) and Ozone Mapping and Profiler Suite (OMPS), to examine the interannual variability of HCHO vertical column density (VCD) during the summer seasons spanning from 2005 to 2019. Our results show that in 2005-2019 summers, wildfires contribute 75-90% of the interannual variability of HCHO VCD over Siberia, Alaska, and Northern Canada, while biogenic emissions and background methane oxidation accounts for ~90% of HCHO interannual variability over Eastern Europe. We find that monthly Solar-induced chlorophyll fluorescence (SIF) from Orbiting Carbon Observatory-2 (OCO-2), an efficient proxy for plant photosynthesis, shows a good linear relationship (R=0.6-0.7) with modelled biogenic HCHO column ($dVCD_{Bio,GC}$) in Eastern Europe, Siberia, Alaska and Northern Canada, indicating the coupling between SIF and biogenic VOC emissions over the four domains on a monthly scale. In Alaska, Siberia and Northern Canada, SIF and $dVCD_{Bio,GC}$ both show a relatively lower interannual variabilities (SIF: CV=1-9%, $dVCD_{Bio,GC}$: CV=1-2%. CV: Coefficient of Variation) comparing to wildfire-induced HCHO (CV=8-13%), suggesting that the high interannual variabilities of OMI HCHO VCD (CV=10-16%) in these domains are likely driven by wildfires instead of biogenic emissions.

"

**Introduction:**

Line 29-30: Consider providing a foundational and more general reference to support this statement, rather than solely referencing works from your co-authors.

We change this sentence to "VOCs are main precursors of tropospheric ozone and secondary organic aerosols, strongly impacting air quality and climate (Atkinson, 2000; Kroll and Seinfeld, 2008; Mao et al., 2018; Zheng et al., 2020). "

Line 43: Avoid just using words such as "large". Provide numbers.

We change this sentence to "During wildfire seasons, pyrogenic emission is the secondary important controlling factor of HCHO over the whole Amazon (Zhang et al., 2019) and contributes to 50-72% of HCHO total column in Alaskan summer fire seasons (Zhao et al., 2022)."

Line 49: The word "predominant" is confusing. What do you mean? The most abundant? The most reactive? or both?

We change this sentence to "Several studies suggest monoterpenes to be the most abundant BVOC species in boreal forests over middle and north Europe, and southeastern Siberia (Spirig et al., 2004; Timkovsky et al., 2010; Bäck et al., 2012; Rantala et al., 2015; Juráň et al., 2017; Zhou et al., 2017)."

Line 53: Same for line 49.

We change this sentence to "This BVOC speciation appears to be different in the boreal forests in Alaska, Northern Canada and Eastern Siberia, where isoprene appears to be the most abundant BVOC species (Blake et al., 1992; Timkovsky et al., 2010; Zhao et al., 2022)."

Line 53-59: The connection or relevance of these points is unclear. Could you clarify the relationship?

The points may not be well connected, so we just keep the point that based on previous measurements.

We change this sentence to "BVOC measurements in tundra systems show a very strong positive temperature dependence for isoprene fluxes, over Greenland (Vedel-Petersen et al., 2015; Kramshøj et al., 2016; Lindwall et al., 2016a), northern Sweden (Faubert et al., 2010; Tang et al., 2016) and the Alaskan North Slope (Potosnak et al., 2013; Angot et al., 2020; Selimovic et al., 2022)."

Line 60: Given the focus on formaldehyde in the preceding and following lines, I guess you want to say the "Wildfire is another important source of formaldehyde"? If so, ensure each word has impact.

Change this sentence to "Wildfire is another important source of HCHO (Permar et al., 2021)."

Line 63: It seems the biogenic and the background are still the main $CH_2O$ sources but fire may be the driven for interannual trend. Please double check your statements here.

We change the sentence to be "A number of studies have shown positive trend and strong interannual variability of wildfires over Arctic regions in the past few decades (Kelly et al., 2013; Giglio et al., 2013; Descals et al., 2022). Several modelling studies suggest that wildfires can become the main source of HCHO over Alaska (Zhao et al., 2022), Siberia and Canada (Stavrakou et al., 2018)."

Line 70: Why using SIF? Is that because other methods having limits? Please provide more context here.

We reorganize the paragraph to be "
Solar Induced Fluorescence (SIF) could potentially provide additional constraints on biogenic-related HCHO column over northern high latitudes, due to their similar dependence on temperature and light availability (Foster et al., 2014; Zheng et al., 2015). SIF is the re-emission of light by plants as a result of absorbing solar radiation during photosynthesis and is widely used to estimate vegetation productivity and health (Porcar-Castell et al., 2014; Magney et al., 2019). Isotopic labeling studies show that 70-90% of isoprene production is from chloroplasts, directly linked to photosynthesis (Delwiche and Sharkey, 1993; Karl et al., 2002; Affek and Yakir, 2003). As SIF is directly linked to flux-derived Gross Primary Productivity (GPP) and HCHO can be largely explained by isoprene emissions (Zheng et al., 2017), we expect to use SIF as a valuable tool to constrain biogenic emissions from boreal forest at northern high latitudes.

"

Line 80-81: Do readers also need to know pros and cons of OMI and OMPS before understanding your work?

We add the sentence to the beginning of the paragraph "
The new retrievals of HCHO from OMI and OMPS provide a continuous long-term record on a global scale, with improved calibration, updates in spectral fitting and air mass factor calculations (González Abad et al., 2022; Nowlan et al., 2023).
"

**Observations and Model:**

Sect. 2.1: Again, why using OMI and OMPS? What are their pros and cons? How does use these two datasets help each other?

We modify the first paragraph of Sect. 2.1 to "
We use satellite observations of tropospheric HCHO columns from OMI and OMPS to evaluate summertime HCHO variability at northern high latitudes. OMI is a UV/Visible backscatter spectrometer on-board the Aura satellite launched in July 2004, with global daily coverage at an overpass time of 13:30 LT. OMI provides a long-term record of HCHO VCD but is discontinued in 2023. OMPS is the continuation of OMI HCHO measurement over polar region. OMPS is a spectrometer on-board two satellites: NASA/NOAA SUOMI NPP (hereafter SNPP) and NOAA-

20, which were launched in October 2011 and November 2017, respectively. Compared to OMI, OMPS-SNPP is in a relatively lower nadir spatial resolution (OMI: 13×24 km$^2$, OMPS-SNPP: 50×50 km$^2$) (de Graaf et al., 2016; Levelt et al., 2006) but has an improved signal-to-noise ratio (González Abad et al., 2016). OMI and OMPS HCHO products share a similar concept and retrieval approach, so the joint evaluation by the two satellites can examine the consistency between OMI and OMPS and, more importantly, provide capability to study HCHO interannual variability on a decadal timescale. Here we use monthly mean HCHO VCD from OMI HCHO VCD retrieval (OMHCHO Version-4) product (González Abad et al., 2022) during 2005-2019 summertime, and OMPS-SNPP Level 2 HCHO total column V1 product (Nowlan et al., 2023) during 2012-2019 summertime, provided by the Smithsonian Astrophysical Observatory. "

Sect. 2.2:

Line 164-173: Have you done the spin up? Also, please use right reference of GEOS-Chem model. For example, in line 176, I believe van der Werf et al. (2017) is the one for GFED4s. Also you need to cite the GEOS-Chem doi with corresponding version. Please read through the web below in GC website.
https://geos-chem.seas.harvard.edu/geos-chem-narrative
http://wiki.seas.harvard.edu/geos-chem/index.php/GEOS-Chem_versions

We add this sentence to the end of the paragraph "
For all model runs, we use a standard restart file from the GEOS-Chem 1-year benchmark simulation, followed by an additional spinup period of several days to allow adequate representation of HCHO production and loss in the model."

In line 176, we change the sentence to "Biomass burning emissions in our simulation are derived from the Global Fire Emission Database (GFED4.1s) inventory (van der Werf et al., 2017; Randerson et al., 2017)."

In line 170-172, we change the URL to GEOS-Chem doi "GEOS-Chem version 12.7.2 is deployed (doi: 10.5281/zenodo.3701669, last access: August 10, 2022) with an update on cloud chemistry (https://github.com/geoschem/geos-chem/issues/906, last access: August 10, 2022). "

Line 178-179: Please cite the original reference of 3-hourly scale factor. You need to read the paper and give credits to people who contribute to the work.

In line 178-179, we change the sentence to "Emissions on a 3-hour basis are obtained from MODIS satellite observations, which provide information on fire detection and burning area (Mu et al., 2011; van der Werf et al., 2017)."

Line 181-182: I believe either Liu or Permar was working on the western US, which is mostly temperate forest. I do not think they are appropriate to compare to boreal forest. If you want to compare boreal forest, you need to cite paper for EF reports from boreal forest or at least

mention the EF reference for GFED4. Again, you need to give credits to people who contribute to the work.

In line 181-183, we change the sentence to "The GFED4.1s inventory reports the HCHO emission factor of 1.86 g/kg and 2.09 g/kg dry matter for boreal forest and temperate forest fires (Akagi et al., 2011)."

Line 185: Results from online and offline MEGAN computation is significantly different. See the web below. I believe offline and online are both turned on by default, which is recommended by GEOS-Chem team. Would the solely online MEGAN influence your results? How do you resolve it?
http://wiki.seas.harvard.edu/geoschem/index.php/MEGAN_v2.1_plus_Guenther_2012_biogenic _emissions#Note_on_the_use_of _pre-computed_emission_factor_.28EF.29_maps_versus_EF_maps_computed_online

We add a paragraph to discuss the difference in HCHO VCD and BVOC emission due to online versus offline MEGAN in Sect. 2.2: "

We conducted a model sensitivity test to assess the difference in BVOC emissions and HCHO $dVCD_{Bio,GC}$ due to online versus offline MEGAN applications. The results of the tests show that the use of online MEGAN has a modest impact on monthly ISOPe and MONOe (25-53% for ISOPe in Alaska, Northern Canada and Eastern Europe, 53% for ISOPe in Siberia; 17-24% for MONOe across the four domains), and provide a similar, isoprene-dominated BVOC emission regime over Alaska, Central Siberia, Northern Canada and Eastern Europe, comparing to results from using offline MEGAN. The difference in $dVCD_{Bio,GC}$ between using online and offline MEGAN is approximately 13-26%, suggesting a minor impact on $dVCD_{Bio,GC}$ and $VCD_{GC}$ variability over northern high latitudes when using online or offline MEGAN.
"

Line 193: Did you explicitly represent the chemistry for isomers or just use the default lumping chemistry for monoterpenes? If I understand it well, the monoterpenes you used is lumped monoterpenes (no?).

We change the line 194-196 to be "In this work, 'monoterpenes' from model calculation are lumped monoterpenes, including $\alpha$-pinene, $\beta$-pinene, sabinene and carene.
"

Line 195-205: Read through narrative description for more information (web provided above). Using MEGAN as an example, you need to cite Hu et al. (2015) as well. Example of how to cite MEGAN in GEOS-Chem narrative description: Biogenic VOC emissions in GEOS-Chem are from the MEGAN v2.1 inventory of Guenther et al. [2012] as implemented by Hu et al. [2015b].

We change line 185-187 to "BVOC emissions in the study are calculated online (Emission factor maps computed online) using the Model of Emissions of Gases and Aerosols from Nature (MEGAN, v2.1) (Guenther et al., 2006, 2012) as implemented by Hu et al (2015)."

Line 197: I believe the isoprene chemistry is updated by Kelvin Bate. Again, please check the narrative description!

It seems the isoprene chemistry updated by Kelvin Bates is added to GEOS-Chem version 12.8.0 and later (https://wiki.seas.harvard.edu/geos-chem/index.php/GEOS-Chem_versions ). In this work we are using GEOS-Chem 12.7.2 so it is not updated in our model.

Line 210: Say three simulations a series of GEOS-Chem simulations. Be specific of what you are trying to write.

We change line 210-214 to "To examine the influence of different sources on HCHO columns in northern high latitudes, we conducted four GEOS-Chem simulations, as described in Table 1, to separate modelled HCHO total column ($VCD_{GC}$) into three parts, including the background column ($VCD_{0,GC}$), biogenic emission induced column ($dVCD_{Bio,GC}$) as well as wildfire emission induced column ($dVCD_{Fire,GC}$):

"

Line 211: Change ":" into "including" might be better?

Fixed.

Line 215: Is it fair to assume the equation is linear for VOCs can be highly impacted by secondary production? How do you resolve the nonlinearity here and how it would influence your conclusion?

We performed model sensitivity tests to examine the impact of non-linear effect on HCHO VCD constitution assumption. We modify the description in Sect. 2.2: "

To examine the influence of different sources on HCHO columns in northern high latitudes, we conducted four GEOS-Chem simulations, as described in Table 1, to separate modelled HCHO total column ($VCD_{GC}$) into three parts, including the background column ($VCD_{0,GC}$), biogenic emission induced column ($dVCD_{Bio,GC}$) as well as wildfire emission induced column ($dVCD_{Fire,GC}$):

$$VCD_{GC} = VCD_{0,GC} + dVCD_{Bio,GC} + dVCD_{Fire,GC} \quad (3)$$

$VCD_{0,GC}$ is the $VCD_{GC}$ from the GEOS-Chem simulation in which both biogenic and wildfire emissions are turned off. $VCD_{0,GC}$, $dVCD_{Fire,GC}$ and $dVCD_{Bio,GC}$ are derived by Eq. (4a) to (4c):

$$VCD_{0,GC} = VCD_{GC}(BG) \quad (4a)$$
$$dVCD_{Fire,GC} = VCD_{GC}(All) - VCD_{GC}(NF) \quad (4b)$$

$$dVCD_{Bio,GC} = VCD_{GC}(NF) - VCD_{0,GC} \qquad (4c)$$
$$dVCD_{Fire,GC}* = VCD_{GC}(NB) - VCD_{GC}(BG) \qquad (4d)$$
$$dVCD_{Bio,GC}* = VCD_{GC}(All) - VCD_{0,GC}(NB) \qquad (4e)$$

To assess the linearity assumption in Eq. (3), we conducted model sensitivity tests over a one-month period to evaluate the disparity between $VCD_{GC}$ and $VCD_{0,GC} + dVCD_{Fire,GC}* + dVCD_{Bio,GC}*$ (derived from Eq.(4a), (4d) and (4e)). The difference between these two terms is less than 14% in northern high latitudes, suggesting a minor importance of the non-linear effect in this area.

**Table 1. Configuration of GEOS-Chem global simulations in this study**

| Simulations | Biogenic emission | Wildfire |
|---|---|---|
| Biogenic + wildfire + Background (All) | On | On |
| Background (BG) | Off | Off |
| Biogenic + Background (NF) | On | Off |
| Wildfire + Background (NB) | Off | On |

"

Line 239-240: Again, is the assumption fair? See above comments for line 215 instead.

Fixed above.

**OMI/OMPS Evaluation with GEOS-Chem HCHO VCD**

Line 250-253: I did not fully read Nowlan et al. (2023) but are you suggesting the correction in that paper does not fix the "high-latitudes bias"?

Yes. During the OMPS HCHO retrieval, a correction is applied to account for high-latitude biases, but some biases may still persist at high latitudes. The lower values in OMPS-SNPP can also be due to some other reasons that mentioned in the line 250-253.

Line 255-258: What's the satellite retrievals uncertainty here? Try to be specific here.

Change the sentence to be "The model-satellite discrepancies in wildfire areas can be in part due to model underestimates of VOC emissions and HCHO production from wildfire plumes (Jin et al., 2023), and in part due to the uncertainties in air mass factor calculation for satellite HCHO retrievals in the presence of wildfire smokes (Jung et al., 2019)."

Line 258-259: What is the "biases in both model and satellites"? Try to be specific rather than using vague words. Figure 2: Is the carbon emissions from emission inventories?

Change line 258-259 to be "The model-satellite discrepancies outside wildfire areas could be also due to model underestimates of oxygenated VOCs (OVOCs), biogenic VOC emissions and biases in satellite HCHO retrieval products. For example, Selimovic et al (2022) found that

GEOS-Chem underestimates OVOCs, including HCHO, by a factor of 3-12 at Toolik Field Station in North Alaska. Stavrakou et al (2015) show model underestimations of biogenic isoprene emission and wildfire emissions over Eastern Europe and Alaska. Recent studies suggest that TROPOMI HCHO retrieval may have a positive bias under low HCHO conditions (Vigouroux et al., 2020). OMPS-SNPP HCHO shows a similar positive bias at clean sites, but has a closer agreement with FTIR HCHO columns at polluted sites (Nowlan et al., 2023; Kwon et al., 2023).”

Add this sentence to Figure 2 caption: “
*Wildfire emissions are calculated from GFED4.1s inventory; biogenic VOC emissions are calculated by MEGAN2.1 model.*
”

Line 312: What is Mann-Kendall test?

Add a sentence before the line 312 sentence: “We use Mann-Kendall test, a non-parametric statistical test used to detect trends in time series data, to test the significance of the trend of monthly HCHO $VCD_{GC}$ time series over a specific domain (Gilbert, 1987).”

Line 315-320: What cause the inconsistency here? Any implications we can gain here?

In Bauwens 2016, the role of wildfire emission in driving HCHO VCD trend is not discussed much.

Change the sentence to “In contrast to Bauwens et al (2016), We find that HCHO $VCD_{GC}$ trend over Siberia is largely driven by the increasing wildfires in recent years, and to a lesser extent by biogenic VOC emissions, highlighting the important role of wildfires on HCHO VCD interannual variability.
”

Line 330: Again, does the vegetation production matter? See my comments for MEGAN.

We add a paragraph to discuss the difference in HCHO VCD and BVOC emission due to online versus offline MEGAN in Sect. 2.2: “

We conducted a model sensitivity test to assess the difference in BVOC emissions and HCHO $dVCD_{Bio,GC}$ due to online versus offline MEGAN applications. The results of the tests show that the use of online MEGAN has a modest impact on monthly $ISOP_e$ and $MONO_e$ (25-53% for $ISOP_e$ in Alaska, Northern Canada and Eastern Europe, 53% for $ISOP_e$ in Siberia; 17-24% for $MONO_e$ across the four domains), and provide a similar, isoprene-dominated BVOC emission regime over Alaska, Central Siberia, Northern Canada and Eastern Europe, comparing to results from using offline MEGAN. The difference in $dVCD_{Bio,GC}$ between using online and offline MEGAN is approximately 13-26%, suggesting a minor impact on $dVCD_{Bio,GC}$ and $VCD_{GC}$ variability over northern high latitudes when using online or offline MEGAN.
”

Line 334-335: There are other potential reasons. NOx level is one for sure. It could also relate to the difference of temperature, photolysis, and oxidants level.

We change the sentence to "This indicates a much lower HCHO production efficiency from isoprene oxidation in northern high latitudes compared to mid-latitude, possibly resulting from the availability of $NO_x$, the difference of temperature, photolysis and oxidants level (Marais et al., 2012; Mao et al., 2013; Li et al., 2016; Wolfe et al., 2016)."

Line 340-345: Might need to weaken the words? You are comparing the campaign-average with your custom model results rather than directly compare to the observation at the exactly time and location if I understand it right.

We change the sentence to "Our model shows comparable isoprene surface mixing ratios with the in-situ measurements along Trans-Siberian Railway within Russian boreal forests (generally <1ppb in our model, and around 0.31–0.48 ppb in the in-situ campaign in Timkovskys et al (2010), both can reach ~4 ppb in Eastern Siberia). Our model also shows comparable monoterpenes surface mixing ratios over Alaskan North Slope (0.009 ppbv in our model and ~0.014 ppbv in Selimovic et al (2022)). "

Line 347: It should be easy to use the recommended veg map in the GEOS-Chem and then figure out if it is the reason (no?).

We believe that the discrepancies between these two models may be related to vegetation map, canopy models and meteorological fields. Changing veg map alone may not reconcile the discrepancy.

We now change the sentence to "
Comparing to Stavrakou et al (2018), our modeled ISOPe over Eastern Europe, Alaska and Northern Canada agrees within 20%, but our modeled MONOe is around 40% lower, likely because we are using online MEGAN, different PFT maps and canopy models (Guenther et al., 2012).
"

Figure 5: Is that seasonally or monthly HCHO? It seems that you only have four data points for each year.

We add this sentence to Figure 5 caption: "*In each year, only the monthly values in May, June, July and August are shown.*"

Line 395: See comments for Line 334-335

We change the sentence to "In Eastern Europe, biogenic emission and background HCHO accounts for the majority of HCHO VCD interannual variability, largely due to the relatively higher surface temperature, stronger photolysis, higher oxidants level and higher availability of $NO_x$ than the other three domains."

Line 397: I might not be familiar with such work but how are BVOC and methane oxidation sensitive to varying temperature?

We modify the sentence to be: "

In regional scale, BVOC emissions and methane oxidation with hydroxyl radicals (OH) both depend on temperature (Guenther et al., 2012; Holmes et al., 2013). In Figure 4d, the surface temperature in Eastern Europe is higher than that in Alaska, Northern Canada and Siberia by 5-7 K, leading to an increase in BVOC emissions and $VCD_{0,GC}$ through methane oxidation.

"

**SIF evaluation on dVCDBio,GC interannual variability**

Line 412-413: What is the specific reason for choosing the Standardized Major Axis (SMA) regression?
Line 415-430: As both SIF and dVCDBio,GC are largely driven by surface temperature, why not just using surface temperature instead? What's the implication of SIF ranges (0-0.25)? What's the benefit of using SIF rather than other proxy?

Because SIF and $dVCD_{Bio,GC}$ are two variables that both related to biogenic isoprene emission, but not directly depend on each other, by using the SMA regression, we can minimize the sum of squares of the residuals in both SIF and $dVCD_{Bio,GC}$.

We rewrite the Sect. 5 'SIF evaluation on $dVCD_{Bio,GC}$ interannual variability' to: "

In Figure 6a to 6d, we find a good linear relationship (R=0.6-0.7) between OCO-2 monthly SIF and $dVCD_{Bio,GC}$ at Alaska, Siberia, Northern Canada and Eastern Europe. Foster et al (2014) show a high linear correlation between seasonal variation of satellite HCHO column (fire free) and GPP in northern high latitudes. This is consistent with our finding over most continental areas in northern high latitudes (Figure S2), since SIF is a widely used proxy of GPP (Frankenberg et al., 2011). In Figure 6g to 6j, SIF and ISOPe show a linear relationship when SIF is within 0-0.25 $Wm^{-2}\mu m^{-1}sr^{-1}$ but tend to decouple when SIF > 0.25 $Wm^{-2}\mu m^{-1}sr^{-1}$, possibly due to the different temperature optimums of isoprene emission and photosynthesis (Harrison et al., 2013; Zheng et al., 2015).

Despite the difference in distribution of vegetation types, the $dVCD_{Bio,GC}$-SIF slope is homogeneous over Siberia, Northern Canada and Eastern Europe (slope=0.28-0.45, unit:$10^{16}$ molecules $cm^{-2}$ per $Wm^{-2}\mu m^{-1}sr^{-1}$), suggesting SIF as a tool to understand biogenic HCHO variability in these regions. The $dVCD_{Bio,GC}$-SIF slope in Alaska is 3-5 times lower than other three domains, which warrants further investigation. In contrast to high latitudes, we find that both ISOPe:SIF slope and $dVCD_{Bio,GC}$:SIF slope are a factor of 2-10 times higher in Southeast

US and Amazon (Figure 6e-6f, 6k-6l) than in northern high latitudes, indicating that the $dVCD_{Bio,GC}$-SIF slope over northern high latitudes and lower latitudes could be very different.

SIF offers an independent evaluation on the interannual variability of HCHO $dVCD_{Bio,GC}$. As SIF showing a linear relationship with $dVCD_{Bio,GC}$ in northern high latitudes (Figure 6a to 6d), it is reasonable to infer from Figure 4 that the low interannual variability shown in SIF (CV=1-9%) is expected for $dVCD_{Bio,GC}$ (CV=1-2%) in Alaska, Siberia and Northern Canada. In contrast, we find that $dVCD_{Fire,GC}$ has a much weaker correlation with SIF (Figure S2c) and shows a higher interannual variability (CV=8-13%). As wildfire emission is highly correlated ($R^2$=78-89%) with OMI HCHO VCD over Northern Canada, Siberia, and Alaska (Figure 3), the high interannual variabilities of OMI HCHO VCD (CV=10-16%) in these domains are likely driven by wildfires instead of biogenic emissions.

"

**Conclusions and discussions**

Line 462: Jin 2021 highlighted the importance of OH in HCHO production. Liu and Permar reported the observed fire ER/EF. How do they suggest underestimated HCHO in wildfire emission?

We modify the discussion in the Sect. 6: "Our modeled HCHO VCD can be biased low, due to large underestimate of HCHO production and emission factor in wildfire smokes. Previous in-situ campaigns show underestimated emission factors of VOCs in GFED4.1s emission inventory for temperate forests in western US (Liu et al., 2017; Permar et al., 2021), while the bias in VOC emission factor in boreal forest wildfires remains unclear. HCHO underestimation can also be due to the missing HCHO secondary production in wildfire-impacted conditions (Liao et al., 2021; Jin et al., 2023). GEOS-Chem is found to underestimate oxygenated VOCs by a factor of 3 to 12 in some Arctic regions, which could contributes to the bias in modelled HCHO in northern high latitudes (Selimovic et al., 2022). More measurements in Arctic region are needed to reconcile the model-observation discrepancies.

"

---

## Author Comment (AC2)

Response to Reviewer #1

We are grateful to the reviewer for the valuable comments that facilitate the important improvements of the original manuscript. We list the point-by-point responses below. The reviewer's comments are marked black and our responses are marked dark blue. Line numbers refer to the discussion paper egusphere-2023-1431.

It is a very interesting article and an alert for future impacts from global warming that will certainly contribute to HCHO emissions.

However, I am not familiar with the methodology used to acquire data.

The OMPS-SNPP, OMI and OCO-2 satellite data are acquired from public online archive or contacting the authors. The GEOS-Chem simulation is based on GEOS-Chem codes acquired online from GEOS-Chem github. We describe the code and data availability after Sect. 6 (Conclusions and discussions):

"**Code and data availability.**
The OMPS-SNPP HCHO L2 V1 product is available at https://disc.gsfc.nasa.gov/datasets/OMPS_NPP_NMHCHO_L2_1/summary (González Abad, 2022). The OMI HCHO L2 product is available at https://waps.cfa.harvard.edu/sao_atmos/data/omi_hcho/OMI-HCHO-L2/ . The OCO-2 SIF is available at https://daac.ornl.gov/cgi-bin/dsviewer.pl?ds_id=1863 (Yu et al., 2021). Data used in this work is available at https://doi.org/10.6084/m9.figshare.23599566.v1 (Zhao, 2023a). Data processing and plotting codes are available at https://doi.org/10.5281/zenodo.8094844 (Zhao, 2023b). The GEOS-Chem model is publicly available at: https://doi.org/10.5281/zenodo.3701669 (GEOS-Chem, 2020).

"

Abstract must contain quantitative results.

Lines 30-32 could use traditional references for HCHO formation, as those from Atkinson, Seinfeld, Pitts and Carter.

Rewrite abstract to be: "
The northern high latitudes (50-90°N, mostly including boreal forest and tundra ecosystem) has been undergoing rapid climate and ecological changes over recent decades, leading to significant variations in Volatile Organic Compounds (VOCs) emissions from biogenic and biomass burning sources. Formaldehyde (HCHO) is an indicator of VOC emissions, but the interannual variability of HCHO and its main drivers over the region remain unclear. In this study, we use the GEOS-Chem chemical transport model and satellite retrievals from Ozone Monitoring Instrument (OMI) and Ozone Mapping and Profiler Suite (OMPS), to examine the interannual variability of HCHO vertical column density (VCD) during the summer seasons spanning from 2005 to 2019. Our results show that in 2005-2019 summers, wildfires contribute 75-90% of the interannual variability of HCHO VCD over Siberia, Alaska, and Northern Canada, while

biogenic emissions and background methane oxidation accounts for ~90% of HCHO interannual variability over Eastern Europe. We find that monthly Solar-induced chlorophyll fluorescence (SIF) from Orbiting Carbon Observatory-2 (OCO-2), an efficient proxy for plant photosynthesis, shows a good linear relationship (R=0.6-0.7) with modelled biogenic HCHO column (dVCD$_{Bio,GC}$) in Eastern Europe, Siberia, Alaska and Northern Canada, indicating the coupling between SIF and biogenic VOC emissions over the four domains on a monthly scale. In Alaska, Siberia and Northern Canada, SIF and dVCD$_{Bio,GC}$ both show a relatively lower interannual variabilities (SIF: CV=1-9%, dVCD$_{Bio,GC}$: CV=1-2%. CV: Coefficient of Variation) comparing to wildfire-induced HCHO (CV=8-13%), suggesting that the high interannual variabilities of OMI HCHO VCD (CV=10-16%) in these domains are likely driven by wildfires instead of biogenic emissions.

"

We change the line 30-32 to be "HCHO is mainly produced from atmospheric VOC oxidation with a short photochemical lifetime on the order of hours, serving as an indicator of non-methane VOC (NMVOC) emissions and photochemical processes (Finlayson-Pitts and Pitts, 1986; Carter, 1994; Atkinson, 1997; Fu et al., 2007; Millet et al., 2008; Seinfeld and Pandis, 2016)."

It is not possible to use other molecules as wildfires tracers as carbon monoxide, and other type of molecules to help to understand the results as COS?

In this work, since COS is not in the standard simulation, it has not been studied yet.

After reorganization the paragraphs, this figure is now Figure 3, and is mainly discussed in Sect.4. We updated this figure to show the contribution of wildfire CO emission in the wildfire carbon emission.

[Figure]

*Figure 3.* *Timeseries of HCHO VCD, biogenic and wildfire emissions over (a) Alaska, (b) Siberia, (c) Northern Canada and (d) Eastern Europe, May 1-August 31, 2005-2019. The blue lines are monthly HCHO VCD from reprocessed OMI, cyan lines are from reprocessed OMPS-SNPP, grey lines are from GEOS-Chem. Red and black bars are area-normalized wildfire CO and VOC emissions during the summer of each year; green bars are area-normalized biogenic VOC emissions. Wildfire emissions are calculated from GFED4.1s inventory; biogenic VOC emissions are calculated by MEGAN2.1 model. Pink shade indicates high HCHO VCD years (definition see Sect. 2.2 and Table S1). The $R^2$ between reprocessed OMI HCHO VCD and biogenic VOC emission (green) / wildfire VOC emission (black) is shown at top right of each panel.*

We change the paragraph in Sect. 4 describing Figure 3 to be: "Figure 3 shows that in Alaska, Northern Canada and Siberia, high HCHO years are often associated with strong wildfire VOC emissions ($R^2$=0.78-0.89) and to a lesser extent associated with biogenic VOC emissions ($R^2$=0.21-0.47). The interannual variability of wildfire VOC emission is further supported by CO emissions from both GFED4 and satellite-based estimation (Yurganov and Rakitin, 2022). The high correlation between OMI HCHO VCD and GFED wildfire VOC emissions in Alaska, Siberia and Northern Canada indicates a strong wildfire impact on interannual variabilities of HCHO VCD in these domains. In Eastern Europe, high HCHO years are associated with large biogenic emissions (With wildfire VOC emissions: $R^2$=0.51; With biogenic emissions: $R^2$=0.72), indicating the important role of biogenic emission in interannual variability of HCHO in Eastern Europe.

"

The results can be extrapolated to South Pole?

According to Riedel et al 1999, HCHO in the Antarctic region is mainly from methane oxidation with OH radicals, with other possible yet unknown HCHO sources. It can be hard to extrapolate our results to the South Pole.

In the introduction, we add the reference about HCHO variability in Antarctic region "
Several studies suggest that biogenic VOC emissions are largely responsible for interannual variabilities of HCHO on a global scale (Palmer et al., 2001; De Smedt et al., 2008; González Abad et al., 2015; De Smedt et al., 2018). Stavrakou et al. (2009) attributes Biogenic VOCs (BVOCs) emissions as the predominant source of global HCHO columns, in which isoprene alone contributes to 30% of global HCHO. Isoprene emissions were also found to be the major driver of HCHO interannual variability (Bauwens et al., 2016; Stavrakou et al., 2018; Morfopoulos et al., 2022). During wildfire seasons, pyrogenic emission is the secondary important controlling factor of HCHO over the whole Amazon (Zhang et al., 2019) and contributes to 50-72% of HCHO total column in Alaskan summer fire seasons (Zhao et al., 2022). Over Antarctic region, HCHO is produced mainly from methane oxidation with hydroxyl radicals (OH), with possible unknown HCHO sources and long-range transport (Riedel et al., 1999). The interannual variability of HCHO over this region is still unclear.
"

---

## Referee Report (RR1)

The authors have sufficiently addressed all prior comments, and the manuscript is generally improved by the revisions. The manuscript conclusions are well supported and clearly stated. I suggest publication as-is.